# Train for Truth, Keep the Skills:
# Binary Retrieval-Augmented Reward Mitigates Hallucinations

Tong Chen [1]   Akari Asai [2 3]   Luke Zettlemoyer [1]   Hannaneh Hajishirzi [1 2]   Faeze Brahman [2]

## Abstract

Modern post-trained language models are increasingly capable, but remain prone to extrinsic hallucinations. We target the utility degradation issue that prior hallucination-reduction methods often struggle to avoid, and propose online RL with Binary Retrieval-Augmented Reward (Binary RAR) to reduce hallucinations while preserving general capabilities. Binary RAR assigns a reward of 1 if a response contains no factual contradictions with retrieved evidence, and 0 otherwise. We theoretically show that this method reduces the probability of error-containing responses while preserving the distribution of error-free responses. This helps preserve the model's capabilities, whereas other methods often degrade them. We evaluate Binary RAR on multiple widely used models. On Qwen3-8B, it reduces long-form hallucination rates by 39.3% and short-form hallucination rates by 54.4%, outperforming supervised learning and preference optimization baselines. Our error analysis shows that continuous factuality rewards (e.g., VeriScore) can be exploited via reward hacking by producing fewer or more generic claims, whereas Binary RAR is more robust and better preserves general capabilities, including instruction following, math, and coding.

## 1. Introduction

Large language models (LMs) demonstrate strong capabilities, but their widespread use is limited by a key reliability issue: extrinsic hallucination, where a model generates plausible yet factually incorrect information (Kalai et al., 2025; Li et al., 2024a). This is becoming more concerning because some recent state-of-the-art reasoning models show higher hallucination rates (Yao et al., 2025; Song et al., 2025).

A common post-training practice is to maximize accuracy on verifiable tasks, which can reward guessing when the model is uncertain (Lambert et al., 2025; Shao et al., 2024). At the other extreme, training can push models to always express uncertainty to avoid errors, which reduces usefulness (Kalai et al., 2025). The goal is to reduce hallucination without sliding toward either extreme, but prior hallucination-reduction methods often still cause utility degradation (Chen et al., 2025). As shown in Table 1, a handful of works apply reinforcement learning (RL) to short-form question answering with ground-truth labels. In the more general long-form setting, where no unique answer exists, how to apply RL while preserving utility remains open, and a holistic analysis of utility degradation is lacking. One line of work measures factuality with continuous scores from claim extraction and per-claim verification (e.g., VeriScore; Song et al., 2024) and applies RL with those scores. While fine-grained, this feedback is easy to exploit: a model can raise the score by changing style or structure, leading to the utility degradation. Some approaches add extra rewards, such as an LM-judge reward (Lin et al., 2024) or a detail-oriented reward (Chen et al., 2025), but these additions complicate training and can add new style bias, leaving utility degradation unresolved.

In this paper, we address the hallucination–utility trade-off with *online* RL using **Binary Retrieval-augmented Reward** (**RAR**; Figure 1 left). We use a binary signal $r \in \{0, 1\}$, where $r = 0$ if any part of the output contradicts the retrieved documents and $r = 1$ otherwise. During training, we retrieve evidence chunks from the web documents and score the model's response by comparing it against this evidence using an LM verifier. Our approach has several advantages. First, we apply continual RL on top of a fully post-trained model rather than training accuracy from scratch, which decouples hallucination reduction from capability acquisition and keeps the training objective simple. Second, the specific design of the reward, contradiction-only, response-level, and binary, is what avoids utility degradation (§6.2). Third, KL regularization alone is sufficient to stabilize training, with no auxiliary judge or detail reward as required by prior work (§3.4).

[1]University of Washington [2]Allen Institute for AI (Ai2) [3]Carnegie Mellon University. Correspondence to: Tong Chen <chentong@cs.washington.edu>.

*Proceedings of the 43rd International Conference on Machine Learning*, Seoul, South Korea. PMLR 306, 2026. Copyright 2026 by the author(s).

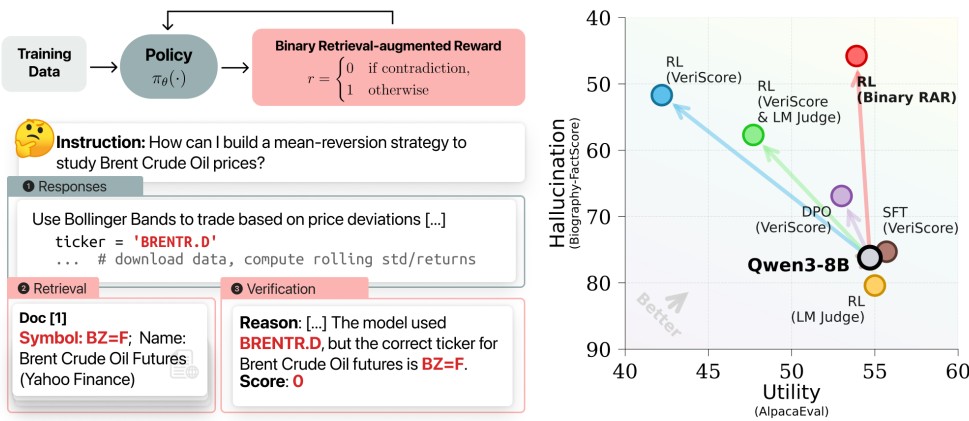

*Figure 1.* Overview of Binary Retrieval-Augmented Reward (Binary RAR). **Left**: Reinforcement learning with Binary RAR assigns a binary reward based on retrieval-verified factual correctness. **Right**: Binary RAR achieves the best hallucination–utility trade-off among all post-training baselines.

We evaluate binary RAR on ten diverse benchmarks to jointly test hallucination reduction and utility preservation (across coding, math, and instruction-following). We compare against prior supervised SFT and DPO baselines, as well as concurrent continuous dense-reward methods, across widely used LMs (Qwen3 and Tulu3). As shown in Figure 1 (right), for long-form generation on Qwen3-8B, Binary RAR reduces the hallucination rate from 76.2 to 45.8, outperforming DPO (66.9) and RL with continuous rewards in a concurrent work (51.7; Chen et al., 2025). Furthermore, even though we never train on explicit abstention labels, penalizing factual errors during RL increases the probability of abstaining when the model is uncertain, reducing the hallucination rate from 60.6 to 27.6 on short-form question answering tasks. This improvement comes with minimal utility loss: the ALPACAEVAL (Dubois et al., 2024) score remains largely stable (-1.4%), whereas VeriScore baselines degrade substantially (-22.8%). These gains are consistent across scales and model families: Binary RAR remains effective on Qwen3-4B and Tulu3-8B.

Our analysis indicates that these factuality gains do not come from making the model less informative or forgetting knowledge. In long-form generation, the trained model produces nearly the same number of correct claims while substantially reducing false claims, so the improvement is mainly higher precision rather than reduced detail. In short-form question answering, when we explicitly prompt the model to avoid abstention and make its best guess, it maintains similar accuracy, suggesting that training does not erase useful knowledge. Our analysis attributes the strong utility preservation to two key factors. First, the reward design reduces reward hacking: response-level verification checks the whole answer, and the binary signal changes only when the verifier decision flips, which weakens incentives for style-driven shortcuts. Second, KL regularization limits its drift from the reference model. In fact, our theoretical

result shows that the KL-regularized optimum simply down-weights error-containing responses while leaving error-free responses unchanged up to normalization.

## 2. Related Work

**Reducing hallucination via post-training.** Prior work explores mitigation via inference-time interventions, such as retrieval-augmented generation (Asai et al., 2024), prompting techniques (Ji et al., 2023b), and decoding algorithms (Chuang et al., 2024). In this work, we focus on training-based approaches to mitigate hallucination. Some studies show that supervised fine-tuning (SFT) can improve factuality by avoiding training on knowledge that the model has not already assimilated during pre-training, as fine-tuning on unfamiliar knowledge can increase the propensity for hallucination (Newman et al., 2025; Zhang et al., 2024). Similarly, Direct Preference Optimization (DPO) trains the model to prefer more factual responses over less factual ones (Tian et al., 2024; Lin et al., 2024). This is often achieved by generating response pairs where preferences are determined by continuous factuality assessment scores. In addition, several prior studies apply on-policy RL only to short-form question answering, where rewards are easier to define and do not extend naturally to long-form generation (Xu et al., 2024; Wei et al., 2025). Concurrent with this work, Chen et al. (2025) combines offline learning (SFT, DPO) with online RL to enhance base LMs' factuality using a continuous factuality signal (i.e., VeriScore).

**Reward hacking in RL for factuality.** A recurring challenge in RL-based factuality training is reward hacking: the model learns shortcuts that raise the training reward but reduce real answer quality. In our setting, the most common failure modes are over-abstention and degraded long-form responses (e.g., overly short or generic). Table 1

| Paper | Training Task | Objective | Avoiding Over-abstention / Degradation |
|---|---|---|---|
| RLKF (Xu et al., 2024) | Short-form | PPO: Reward model | Reward: Correct > Abstain > Wrong |
| TruthRL (Wei et al., 2025) | Short-form | GRPO: Ternary reward | Reward: +1 (correct), 0 (abstain), -1 (wrong) |
| FactTune (Tian et al., 2024) | Long-form | DPO: FactScore | Not Studied |
| FLAME (Lin et al., 2024) | Long-form | DPO: FactScore + LM-judge | Preference: LM-judge |
| Reason for Factuality (Chen et al., 2025) | Long-form | GRPO: FactScore + LM-judge + Detail-oriented | Reward: LM-judge / Detail-oriented |
| **Binary RAR** | **Long-form** | **GRPO: Binary Retrieval-Augmented Reward** | **No additional reward** |

*Table 1.* Related work comparison for hallucination reduction and abstention control. We contrast prior short-form and long-form post-training methods by training objective and the mechanism used to discourage over-abstention. Binary RAR uses a binary retrieval-grounded contradiction reward trained with GRPO, with KL control to preserve answer utility.

summarizes how prior work controls these issues: short-form methods explicitly shape the reward to prefer *correct* answers over *abstain* and *wrong*, while long-form methods often rely on auxiliary preference signals such as LM-judge or detail-oriented rewards to reduce degradation. Binary RAR differs by using a retrieval-grounded *binary* contradiction reward, avoiding extra judge-based objectives while preserving general capability.

## 3. RL with Binary Retrieval-Augmented Reward

This section presents an overview (§3.1), the training objective (§3.2), binary reward with retrieval and verification (§3.3), and discusses how this method avoids utility degradation (§3.4).

### 3.1. Overview

We use *online RL*, where rewards are computed on the model's *own rollouts*. We introduce a *binary retrieval-augmented reward* (**Binary RAR**; Figure 1) that assigns reward 1 if the response contains no factual contradictions with retrieved evidence, and 0 otherwise. This binary design supports both short-form question answering and long-form generation because it does not require a single ground-truth answer; instead, it checks the response against web documents retrieved for the prompt and the response. We optimize a KL-regularized objective to limit drift from a reference model. Overall, the update reduces the probability of contradiction-containing responses while largely preserving responses that are already correct or appropriately abstain.

**Formulation.** Our goal is to reduce hallucination while preserving the general capabilities of a fully trained LM. Let $x$ be an instruction and $y \sim \pi_\theta(\cdot \mid x)$ be the response from model $\pi_\theta$. Let $r_\mathrm{h}(x,y) \in \mathbb{R}$ be the hallucination metric on $x \sim \mathcal{D}_\mathrm{hall}$, and let $r_\mathrm{u}(x,y) \in \mathbb{R}$ be the utility metric on $x \sim \mathcal{D}_\mathrm{util}$. Define $H(\theta) = \mathbb{E}[r_\mathrm{h}(x,y)]$ and $U(\theta) = \mathbb{E}[r_\mathrm{u}(x,y)]$, where the expectation is over $x$ from the corresponding dataset and $y \sim \pi_\theta(\cdot \mid x)$. We prefer a model $\theta_1$ over another $\theta_2$ if $H(\theta_1) \leq H(\theta_2)$ and $U(\theta_1) \geq U(\theta_2)$.

### 3.2. Preliminaries of Reinforcement Learning

The goal of RL is to train a language model to maximize a reward function $r(x,y)$, which assigns a scalar score to the generated response. The objective is formally expressed as:

$$\max_{\pi_\theta} \quad \mathbb{E}_{\substack{x \sim \mathcal{D} \\ y \sim \pi_\theta(\cdot|x)}} \Big[ r(x,y) - \beta \, \mathbb{D}_\mathrm{KL}\big(\pi_\theta(\cdot \mid x) \, \| \, \pi_\mathrm{ref}(\cdot \mid x)\big) \Big], \tag{1}$$

where $\mathcal{D}$ is the prompt dataset, $\mathbb{D}_\mathrm{KL}$ is the Kullback–Leibler (KL) divergence term against a reference model $\pi_\mathrm{ref}$, and $\beta$ controls the strength of the KL penalty.

Several algorithms exist to optimize this objective. Among them, Group Relative Policy Optimization (GRPO; Shao et al. 2024) has become a popular choice for LM post-training due to its stability and computational efficiency (DeepSeek-AI et al., 2025). Specifically, for each prompt $x$, GRPO samples a group of outputs $y_1, ..., y_n$ from the old policy $\pi_\mathrm{old}$ and optimizes the policy model $\pi_\theta$ by maximizing:

$$\mathbb{E}_{\substack{x \sim \mathcal{D} \\ \{y_i\}_{i=1}^n \sim \pi_\mathrm{old}(\cdot|x)}} \left[ \frac{1}{n} \sum_{i=1}^n \frac{1}{|y_i|} \sum_{t=1}^{|y_i|} \min\left( \frac{\pi_\theta(y_i^t \mid y_i^{<t}, x)}{\pi_\mathrm{old}(y_i^t \mid y_i^{<t}, x)} A_i, \right. \right.$$
$$\left. \left. \mathrm{clip}\left( \frac{\pi_\theta(y_i^t \mid y_i^{<t}, x)}{\pi_\mathrm{old}(y_i^t \mid y_i^{<t}, x)}, 1-\epsilon, 1+\epsilon \right) A_i - \beta \, \mathbb{D}_\mathrm{KL}(\pi_\theta \| \pi_\mathrm{ref}) \right], \tag{2}$$

where $\epsilon$ and $\beta$ are hyperparameters, and the advantage $A_i$ and KL regularization $\mathbb{D}_\mathrm{KL}$ are defined as:

$$A_i = \frac{r(x,y_i) - \mathrm{mean}[\, r(x,y_1), ..., r(x,y_n)\,]}{\mathrm{std}[\, r(x,y_1), ..., r(x,y_n)\,]} \tag{3}$$

$$\mathbb{D}_\mathrm{KL}(\pi_\theta \| \pi_\mathrm{ref}) = \frac{\pi_\mathrm{ref}(y_i \mid x)}{\pi_\theta(y_i \mid x)} - \log \frac{\pi_\mathrm{ref}(y_i \mid x)}{\pi_\theta(y_i \mid x)} - 1 \tag{4}$$

We adopt GRPO as the default RL algorithm for our experiments.

### 3.3. Binary Retrieval-Augmented Reward

**Pipeline.** We define the factual correctness of an instruction-response pair $(x,y)$ as the consistency between

the generated content and reliable sources. A pair is considered correct if all information in $y$ is supported by evidence. We introduce a binary retrieval-augmented reward $r(x, y) \in \{0, 1\}$ and use it as a proxy for true factual correctness in the RL training (Figure 1, left):

- **Retrieval.** A datastore $\mathcal{DS} = \{d_i\}_{i=1}^{M}$ consists of reliable documents that are preprocessed, chunked, and indexed by a retriever $R$. To verify factuality, we retrieve the top $k$ relevant documents $C(x, y)$ for each $(x, y)$ pair based on text similarity. These documents serve as evidence for verification.

- **Verification.** To check correctness, an LM verifier takes $(x, y, C(x, y))$ as input and determines whether contradictions exist between the response and the retrieved documents. The verifier focuses solely on contradictions, given the context of $x$. Formally,

$$r(x, y) = \begin{cases} 1 & \text{if no contradictions are found between} \\ & \quad (x, y) \text{ and } C(x, y), \\ 0 & \text{otherwise.} \end{cases}$$
(5)

We then optimize the KL-constrained RL objective (Equation 2) with this binary retrieval-augmented reward. This approach avoids the complexity of continuous reward design and provides a cleaner, less noisy training signal. Prompting details are given in Appendix D.

**Efficiency Consideration.** We pre-cache a small document set $\mathcal{DS}_{\text{cache}}(x)$ for each prompt during dataset preparation, so training-time retrieval draws $C(x, y)$ from the cache instead of the full datastore $\mathcal{DS}$. For verification, we compare the full response $y$ against the retrieved documents $C(x, y)$ in a single LM forward pass, rather than extracting and checking claims one by one. This reduces repeated document processing and gives a $2\times$–$4\times$ throughput improvement (depending on response length) in our setup. See more details in §C.

### 3.4. Understanding Utility Preservation

It is useful to view the KL-regularized RL objective (Equation 1) as the sum of two competing terms: a *reward term* that encourages contradiction-free generations, and a *KL term* that penalizes drifting away from the fully trained reference model.

$$\mathbb{E}\Big[ \underbrace{r(x, y)}_{\text{reward}} - \beta \underbrace{\mathbb{D}_{\text{KL}}(\pi_\theta \parallel \pi_{\text{ref}})}_{\text{KL}} \Big].$$
(6)

The key point of binary RAR is that unnecessary changes to an already-correct response do not increase the reward term, but they do increase the KL divergence term. Therefore, when $\pi_{\text{ref}}$ already produces a contradiction-free answer (either a correct answer, or an appropriate "I don't know" in

short-form settings), the best update is to keep the distribution close to $\pi_{\text{ref}}$ rather than moving probability mass toward abstention.

**Theorem 1.** *The optimal solution $\pi^\star(y \mid x)$ of this KL-regularized objective Equation 1 uniformly downweights every response that contains a factual error by a factor of $\exp(-1/\beta)$ (relative to $\pi_{\text{ref}}$), while leaving every contradiction-free response unchanged, up to a normalization constant. Formally,*

$$\pi^\star(y \mid x) \propto \pi_{\text{ref}}(y \mid x) \times \begin{cases} 1 & \text{if } r(x, y) = 1, \\ \exp(-1/\beta) & \text{if } r(x, y) = 0. \end{cases}$$
(7)

We provide a proof in §F. Equation 7 shows that Binary RAR only applies a uniform penalty to the set of error-containing responses. As a result, among contradiction-free responses (those with $r(x, y) = 1$), their relative probabilities are preserved (they are all scaled by the same factor), so the policy keeps the reference model's preferences within the error-free region. This helps preserve the original capabilities of the initial model $\pi_{\text{ref}}$ while reducing probability mass on error-containing outputs.

## 4. Experimental Setup

### 4.1. Benchmarking the Hallucination–Utility Trade-off

We curate an evaluation suite that includes four datasets for *hallucination evaluation* and ten datasets for *utility evaluation*, spanning math, code, general chat, and instruction following. Our objective is to *minimize hallucination errors while avoiding performance degradation on utility benchmarks* relative to the original LM.

**Hallucination Evaluation** We evaluate hallucination reduction on two task types: long-form generation and short-form question answering. (1) *Long-form generation:* We prompt models on BIOGRAPHY (Min et al., 2023) and WILDHALLUCINATION (Zhao et al., 2024). We want the model to provide the correct information while avoiding factual errors. Following the long-form hallucination metric in OpenAI (2025), the hallucination rate is computed as the proportion of incorrect claims among all extracted atomic claims, which equals $1 -$ precision in FactScore (Min et al., 2023). (2) *Short-form question answering:* We evaluate on POPQA (Mallen et al., 2023) and GPQA (Rein et al., 2024), where each question is associated with a set of ground-truth answers. We instruct the model to output the exact string "I don't know" when it is uncertain (abstention-allowed setting), and we use string matching to detect abstentions. The hallucination rate is the proportion of incorrect test cases.

**Utility Evaluation** We evaluate utility preservation across three categories. (1) *Open-ended generation with LM-judge:*

We use ALPACAEVAL (Dubois et al., 2024) and ARENA-HARD (Li et al., 2025). We find that training for long-form hallucination reduction can noticeably affect these tasks because different rewards can shift writing style. We therefore treat these benchmarks as primary indicators of utility preservation. We compute scores with the official repositories and enable length control to reduce length-related bias. (2) *Short-form QA without abstention:* We re-evaluate POPQA (Mallen et al., 2023) and GPQA (Rein et al., 2024) in an abstention-disabled setting by prompting the model to always produce its best guess. This tests whether training causes knowledge forgetting that is masked by abstention. (3) *Other capabilities:* We include IFEVAL (Zhou et al., 2023) for instruction following; BBH (Suzgun et al., 2023), GSM8K (Cobbe et al., 2021), and MINERVA (Lewkowycz et al., 2022) for reasoning; and HUMANEVAL (Chen et al., 2021) and MBPP (Austin et al., 2021) for code generation. We follow each benchmark's official evaluation protocol. Full details are in (§B).

### 4.2. Training Models and Dataset Curation

We perform continual RL fine-tuning on Qwen3 (4B and 8B; Qwen-Team, 2025) in our main experiments, and we evaluate other base models (Tulu3) in §6.4. We use GRPO algorithm for computational efficiency, although binary RAR is compatible with any RL algorithms. We use Qwen3-32B as the verifier for Binary RAR, prompting it to detect contradictions between the model response and the retrieved documents. We choose Qwen3-32B because it achieves higher verification accuracy than Qwen3-8B (see §A.2).

Curating high-quality and diverse prompts is essential for effective RL training (Kimi-Team et al., 2025). We aim to reduce hallucinations across diverse knowledge domains and instruction types by using natural prompts that reflect realistic user interactions. We build on WildChat (Zhao et al., 2024), a large collection of natural instruction–response pairs from human interactions with OpenAI models. From this dataset, we automatically identify examples whose responses contain verifiable factual content. Concretely, we use the OpenAI `gpt-4.1` model with a detailed classification prompt to select suitable examples (see §D).

### 4.3. Baselines

We compare our method against supervised fine-tuning (SFT), direct preference optimization (DPO), and online RL with some reward signals (Tian et al., 2024; Lin et al., 2024; Chen et al., 2025). For each model, we generate eight responses and evaluate their factuality using the VeriScore pipeline.[1] Specifically, we extract verifiable claims from

---

[1]We do not use SFT or DPO with Binary RAR because, for many prompts, all sampled responses receive the same binary reward. This leaves too few informative training examples.

each response, verify them against pre-cached documents, and compute the percentage of correct claims. For SFT, we fine-tune on the most factual response per prompt. For DPO, we construct preference pairs using the two responses with the largest factuality gap and a length difference below 10%, to prevent "length hacking" (Chen et al., 2025). For RL-based baselines, we consider different reward functions. We first use LM Judge, which rates overall response quality on a 0–10 scale, following common practice (Gunjal et al., 2025). We also test VeriScore (Song et al., 2024) as an RL reward, following concurrent work (Chen et al., 2025).

## 5. Main Results

### 5.1. Results on Hallucination Reduction

Table 2 summarizes hallucination rates across long-form generation and short-form question answering. The base Qwen3-8B model exhibits substantial hallucination, producing 61.9% incorrect claims in long-form generation and 60.6% incorrect answers in short-form QA.

**SFT and DPO Provide Limited Hallucination Reduction.** SFT and DPO applied to responses with high VeriScore yield only modest improvements in factuality. On Qwen3-8B, hallucination reduction is small for both long-form (SFT: -1.0; DPO: -8.5) and short-form (SFT: -0.4; DPO: -3.4) settings. These methods rely on an *offline* dataset collected once with the base model. Consequently, factual errors remain in both SFT labels and DPO preferred sequences even after the model evolves, limiting their effectiveness.

**Binary RAR Outperforms Other Online RL Rewards.** Among all RL-based approaches, Binary RAR delivers the most consistent and substantial reduction in hallucination. On Qwen3-8B, it lowers long-form hallucination from 61.9 to 37.5 (-24.4) and short-form from 60.6 to 27.6 (-33.0), outperforming all baselines. By contrast, RL with the continuous VeriScore reward achieves a moderate hallucination reduction (long-form: -21.3; short-form: -18.3) but shows a significant reduction on ALPACAEVAL and ARENAHARD as shown in Table 3. Alternatively, optimizing for a general LM-judge reward unfortunately increases long-form hallucination (+3.5), suggesting that optimizing for broad instruction-following can conflict with factual accuracy.

### 5.2. Results on General Capabilities Preservation

We evaluate utility preservation on ten benchmarks spanning instruction following, knowledge retention, reasoning, and coding. Binary RAR not only reduces hallucination but also best preserves general capabilities.

**Open-Ended Chat is Sensitive to Hallucination Reduction.** Table 3 reports the win rate on ALPACAEVAL and

| Models | Long-form (Hallucination Rate ↓) | | | Short-form (Hallucination Rate ↓) | | |
|---|---|---|---|---|---|---|
| | BIOGRAPHY | WILDHALLU | AVG | POPQA | GPQA | AVG |
| **Qwen3-8B** | 76.2 | 47.6 | 61.9 | 71.2 | 50.0 | 60.6 |
| + SFT | 75.3 | 46.5 | 60.9 | 70.4 | 50.0 | 60.2 |
| + DPO | 66.9 | 39.8 | 53.4 | 65.2 | 49.1 | 57.2 |
| + RL (LM Judge) | 80.4 | 50.3 | 65.4 | 68.8 | 48.0 | 58.4 |
| + RL (VeriScore) | 51.7 | 29.5 | 40.6 | 43.6 | 41.1 | 42.3 |
| + RL (Binary RAR) | 45.8 | 29.2 | 37.5 | 26.8 | 28.3 | **27.6** |
| **Qwen3-4B** | 81.9 | 50.5 | 66.2 | 82.2 | 55.1 | 68.7 |
| + SFT | 78.9 | 48.7 | 63.8 | 83.8 | 54.7 | 69.2 |
| + DPO | 73.4 | 43.9 | 58.7 | 82.6 | 54.5 | 68.5 |
| + RL (LM Judge) | 82.6 | 53.7 | 68.1 | 80.4 | 54.0 | 67.2 |
| + RL (VeriScore) | 61.1 | 32.6 | 46.9 | 73.0 | 51.3 | 62.2 |
| + RL (Binary RAR) | 46.5 | 28.9 | 37.7 | 46.6 | 37.3 | **41.9** |

*Table 2.* Factuality results comparing different training methods on long-form generation and short-form question answering tasks. We report hallucination rate for both long-form and short-form question answering. Binary RAR achieves the best hallucination reduction.

| Models | ALPACA-EVAL | ARENA-HARD |
|---|---|---|
| **Qwen3-8B** | 54.7 | 18.7 |
| + SFT | 55.7 | 17.4 |
| + DPO | 53.0 | 18.3 |
| + RL (LM Judge) | 55.0 | 18.0 |
| + RL (VeriScore) | 42.2 | 14.9 |
| + RL (Binary RAR) | 53.9 | 17.9 |
| **Qwen3-4B** | 41.7 | 12.6 |
| + SFT | 41.2 | 8.2 |
| + DPO | 39.6 | 11.0 |
| + RL (LM Judge) | 42.3 | 11.5 |
| + RL (VeriScore) | 38.4 | 11.7 |
| + RL (Binary RAR) | 43.0 | 12.5 |

*Table 3.* Utility evaluation on open-ended tasks shows that Binary RAR degrades much less than VeriScore.

| Models | IF | Knowledge | | Reasoning | | | Coding | | |
|---|---|---|---|---|---|---|---|---|---|
| | IFEVAL | POPQA | GPQA | BBH | GSM8K | MINERVA | HUMAN-EVAL | MBPP | AVG |
| **Qwen3-8B** | 87.2 | 20.2 | 48.2 | 62.4 | 92.8 | 80.7 | 83.5 | 67.4 | 67.8 |
| + SFT | 86.9 | 20.4 | 47.9 | 59.4 | 91.6 | 82.0 | 83.8 | 67.0 | 67.4 |
| + DPO | 84.5 | 18.6 | 47.5 | 62.3 | 90.8 | 82.1 | 86.7 | 67.8 | 67.5 |
| + RL (LM Judge) | 82.2 | 19.2 | 52.2 | 63.1 | 88.1 | 77.7 | 83.8 | 66.3 | 66.6 |
| + RL (VeriScore) | 88.7 | 19.6 | 47.7 | 61.4 | 92.2 | 79.0 | 83.4 | 66.9 | 67.4 |
| + RL (Binary RAR) | 85.2 | 20.6 | 48.8 | 66.4 | 93.4 | 82.3 | 86.1 | 67.6 | 68.8 |
| **Qwen3-4B** | 86.1 | 16.4 | 44.2 | 60.9 | 91.1 | 82.8 | 85.5 | 65.7 | 66.6 |
| + SFT | 82.6 | 15.2 | 43.5 | 59.6 | 91.4 | 83.6 | 83.2 | 65.6 | 65.6 |
| + DPO | 81.9 | 15.8 | 44.0 | 63.7 | 90.1 | 82.7 | 85.8 | 66.3 | 66.3 |
| + RL (LM Judge) | 74.3 | 16.0 | 43.5 | 58.1 | 87.0 | 82.1 | 85.9 | 66.2 | 64.1 |
| + RL (VeriScore) | 86.0 | 15.4 | 40.8 | 59.1 | 90.8 | 82.5 | 84.5 | 66.2 | 65.7 |
| + RL (Binary RAR) | 84.7 | 16.4 | 42.6 | 58.5 | 90.7 | 83.8 | 84.6 | 65.0 | 65.8 |

*Table 4.* General capability results on the remaining eight benchmarks: instruction following (IFEVAL), knowledge (POPQA, GPQA), reasoning (BBH, GSM8K, MINERVA), and coding (HUMANEVAL, MBPP). We color each cell based on the relative change compared to the base model, where deeper red indicates larger degradation.

ARENAHARD. We find they are the most sensitive benchmarks to hallucination reduction methods. Both use an LM judge to approximate human preference for long-form outputs, capturing aspects such as relevance, helpfulness, and completeness of the generated responses. When trained with VeriScore-based RL, the model shows substantial performance drops on ALPACAEVAL (54.7→42.2) and ARENAHARD (18.7→14.9). This degradation suggests that continuous rewards such as VeriScore are prone to reward hacking, where the model over-optimizes the proxy signal at the cost of overall response quality. In contrast, RL with Binary RAR preserves scores on these benchmarks, indicating stronger robustness against such overfitting.

**Binary RAR Leads to Calibrated Abstention Behavior.**
In the short-form QA evaluation, we further categorize answers into three types: correct, incorrect, and abstaining, as shown in Figure 2. A consistent emergent behavior is that the model abstains more often when it is likely to be wrong. After Binary RAR training, the Qwen3-8B model's behavior changes substantially: it abstains on 55.2% of POPQA and 27.5% of GPQA questions. The change in accuracy is small, with only a 5% absolute difference on each dataset.

This indicates that the model strategically chooses to abstain when uncertain rather than refusing to answer arbitrarily.

**Factual Knowledge is Well Preserved.** We also evaluate POPQA and GPQA under a no-abstention setting. In our hallucination evaluation, the model is instructed to abstain when it does not know. Here, we instead require the model to always provide its best guess. As shown in Table 4, Binary RAR maintains or slightly improves accuracy (POPQA: 20.2→20.6; GPQA: 48.2→48.8), suggesting that increased abstention reflects better uncertainty calibration rather than loss of factual knowledge.

**Reasoning and Coding Remain Intact.** Table 4 also reports results on several reasoning and coding benchmarks. Across methods, performance changes are minimal. A likely explanation is that our factuality-focused training data has limited overlap with these domains, while math and coding performance is driven mainly by structured reasoning rather than factual recall.

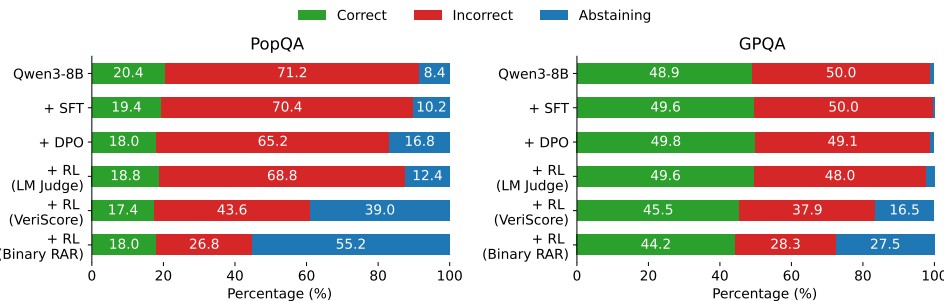

*Figure 2.* Abstention behavior in short-form question answering. Binary RAR leads the model to abstain on uncertain questions rather than producing incorrect answers, preserving accuracy for attempted ones.

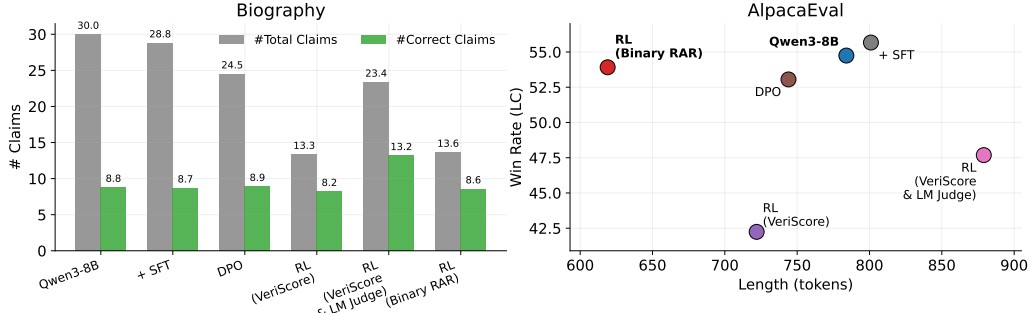

*Figure 3.* Informativeness in long-form generation. **Left**: On BIOGRAPHY, Binary RAR cuts the total number of claims but keeps correct claims nearly the same, showing selective filtering of uncertain content. **Right**: On ALPACAEVAL, Binary RAR gives shorter answers with similar win rates, showing it stays concise without losing quality.

## 6. Analysis

In this section, we analyze the key factors behind utility preservation. We study open-ended generation in depth (§6.1), and examine the reward design (§6.2) and the KL coefficient (§6.3). We also evaluate additional model families (§6.4).

### 6.1. Informativeness in Long-form Generation

Although RL with Binary RAR appears to make model outputs less verbose, a closer examination reveals that the informativeness of correct content remains largely unchanged. Figure 3 (left) shows that on the BIOGRAPHY dataset, the total number of claims decreases from 30.0 to 13.6 after Binary RAR training, yet the number of correct claims remains nearly constant (8.8→8.6). This indicates that the model does not simply drop details or shorten text indiscriminately. Instead, it selectively filters out uncertain statements while preserving confident and factually supported information. In other words, the reduction in hallucination arises from improved selectivity rather than content loss.

A similar pattern holds when examining the length and win rate on ALPACAEVAL. As shown in Figure 3 (right), the Binary RAR model generates shorter responses but maintains a comparable win rate. Its length-controlled win rate (54.7→53.9) and vanilla win rate (59.4→59.3) remain

mostly unchanged. This suggests that Binary RAR learns to produce more concise yet equally effective outputs and avoids unnecessary verbosity while maintaining the same level of perceived helpfulness and informativeness.

We provide additional case studies and discussion in §A.1. As a continuous reward, VeriScore can incentivize behaviors that conflict with human preferences. In particular, models can exploit VeriScore by (1) generating irrelevant but factually correct information and (2) producing high-level, trivially true statements instead of informative details.

### 6.2. Analysis on Reward Design and Reward Hacking

We also evaluate three alternative reward designs that use either claim-level verification or a continuous score. Although they reduce hallucination, all remain vulnerable to reward hacking in practice (Figure 4). (1) *Binary VeriScore*: Thresholding VeriScore at 0.5 converts the continuous score into a binary reward, but it remains sensitive to output style and leads to utility degradation. (2) *Conflict-only VeriScore*: Using the fraction of non-contradictory claims (rather than supported claims) reduces noise from retrieval failures, because irrelevant evidence tends to yield similar rewards across responses. However, the model exploits this by producing less relevant but easier-to-justify statements, which lowers ALPACAEVAL. (3) *Rating-based RAR*: Replacing the binary decision with a 0–10 factuality rating from the

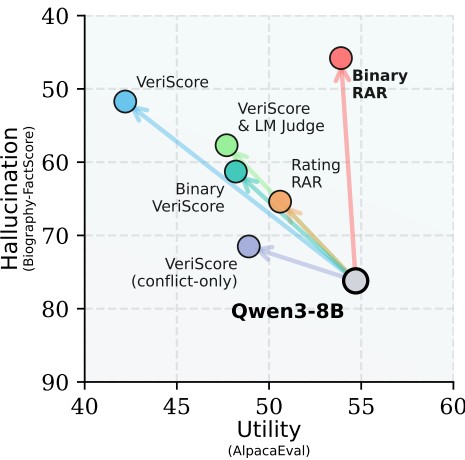

*Figure 4.* Comparison of reward designs. Binary RAR outperforms both continuous (e.g., *Rating RAR*) and claim-extraction-based (e.g., *Binary VeriScore*, *Conflict-only VeriScore*) rewards.

same LM verifier removes dependence on claim extraction, but the model then exploits the verifier's preferences for certain response styles. Overall, these results suggest that Binary RAR benefits from (i) evaluating the response as a whole and (ii) using a binary correctness reward that is comparatively robust to style-driven reward hacking.

Prior work suggests that adding an additional utility reward (e.g., an LM-judge reward) can mitigate utility degradation. However, we find that its effect is limited (*VeriScore & LM Judge* in Figure 4). We speculate that this is because Qwen3-8B has already been carefully trained with human feedback, and an LM-judge reward remains vulnerable to reward hacking.

### 6.3. Effect of KL Coefficient and Early Stopping

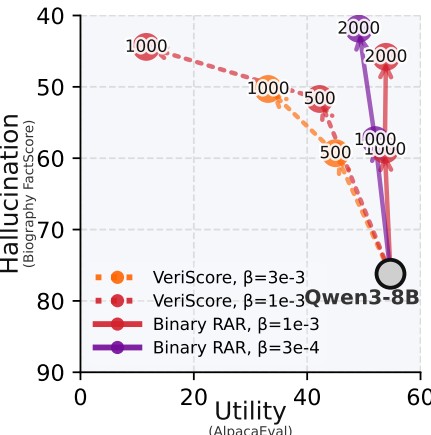

*Figure 5.* Hallucination-utility trade-off across KL coefficients. Each dot is a checkpoint, and its label indicates the number of gradient updates.

Figure 5 sweeps the KL coefficient for two reward choices (Binary RAR and VeriScore) and evaluates intermediate checkpoints along each training run (Binary RAR: 1K and

2K steps; VeriScore: 500 and 1K steps). This lets us separate two controls: training steps and the KL coefficient. Early stopping is well-motivated because the runs show a consistent trajectory: hallucination tends to decrease as training proceeds, while utility also tends to decrease, and once utility drops substantially it rarely recovers. After a checkpoint has already degraded in ALPACAEVAL, training longer typically makes it worse, so there is little value in extending those runs for model selection.

The KL coefficient controls how much the policy is allowed to drift from the initial checkpoint. Binary RAR is relatively insensitive to this choice: varying $\beta$ changes results only modestly. This supports our discussion in §3.4: with a binary reward, the KL term mainly sets how strongly we downweight the probability mass of error-containing responses, while leaving the relative probabilities among contradiction-free responses largely unchanged (up to normalization). This helps preserve the original capabilities of the base model. VeriScore is more sensitive to $\beta$ because the continuous reward interacts more with stylistic and claim-level degrees of freedom; a stronger KL constraint can partially prevent drift and reduce degradation, but it also limits factuality gains, creating a sharper tuning trade-off than Binary RAR.

### 6.4. Other Base Models

To test whether Binary RAR transfers to other model families, we apply Binary RAR to Tulu3-8B (built on Llama-3.1-8B and trained without chain-of-thought). Table 5 shows that it reduces the hallucination ratio on Biography from 68.1 to 60.1 and on WildHallu from 43.4 to 39.3, while utility remains nearly unchanged (AlpacaEval 24.3 → 23.4). This shows that Binary RAR improves factuality without harming a strong, preference-tuned base.

## 7. Conclusion

We study the hallucination–utility trade-off and propose online reinforcement learning (RL) with a binary retrieval-augmented reward (Binary RAR) to reduce extrinsic hallucinations without degrading general capabilities. We theoretically show that this objective decreases probability mass on error-containing responses while preserving the distribution over error-free responses from the base model. Empirically, Binary RAR consistently reduces hallucination rates while keeping utility close to the base model. Further analysis shows that its robustness comes from (1) the binary reward, which is less sensitive to style and harder to exploit, and (2) response-level verification, which avoids brittle claim extraction and reduces reward hacking.

## Impact Statement

This research aims to mitigate extrinsic hallucinations in language models to improve reliability in real-world use. By reducing factual contradictions, it can lower the risk of producing and spreading misinformation. Our method uses publicly available data and a retrieval-and-verification pipeline to score outputs, and it is not designed to introduce new biases. However, retrieval or verifier errors may still affect some topics more than others, so careful source selection and monitoring remain important. In the long term, we hope this line of work supports language models that are both helpful and consistently trustworthy in high-stakes, long-form settings.

## Acknowledgment

We thank members of UW NLP, UW ML, and the AllenNLP team at Ai2 for their helpful feedback. We also thank Luca Soldaini, Teng Xiao, Pradeep Dasigi, Rulin Shao, Zhiyuan Zeng, Scott Geng, Hamish Ivison, Xinran Zhao, Jacob Morrison, and Pang Wei Koh for valuable discussions throughout the project.

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

# A. Additional Analysis

## A.1. Qualitative Analysis

To better understand the impact of RL training with Binary RAR, VeriScore, and the LM Judge, we present a qualitative analysis of the reward signals and the resulting fine-tuned models.

**LM Judge Alone Provides Limited Factuality Assessment.** Figure 6 presents two responses to the same instruction along with their evaluations from all three reward models. While the first response contains a factual error and the second is entirely correct, all three rewards appropriately assign lower scores to the erroneous response. However, the LM Judge prioritizes detailed elaboration over factual correctness. When the factual error in the first response is corrected, the Judge only increases its score by 0.1, suggesting that it values comprehensive coverage more than accuracy. This limitation highlights why the LM Judge alone is insufficient for ensuring factuality.

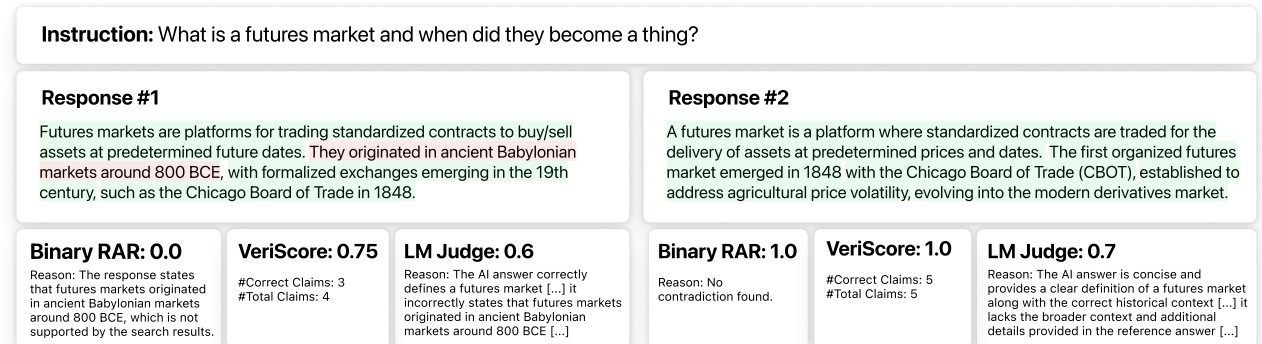

*Figure 6.* Comparing Binary RAR, VeriScore, and LM Judge on LM responses. Each reward provides a score and reason, with green and red highlights marking correct and incorrect facts; Binary RAR penalizes factual errors most reliably.

**VeriScore is Vulnerable to Reward Hacking.** As a continuous reward function, VeriScore can incentivize behaviors that conflict with human preferences. Specifically, models can exploit VeriScore in two ways: (1) by generating irrelevant information that is factually correct, and (2) by producing high-level, trivially true statements rather than informative details. Figure 7 illustrates this behavior through examples from models trained with Binary RAR versus VeriScore. The model trained with Binary RAR produces well-structured outputs that contain many details, while the VeriScore-trained model tends to generate more superficial, higher-level descriptions. This demonstrates that continuous reward signals, while well-intentioned, can lead to undesirable failure modes.

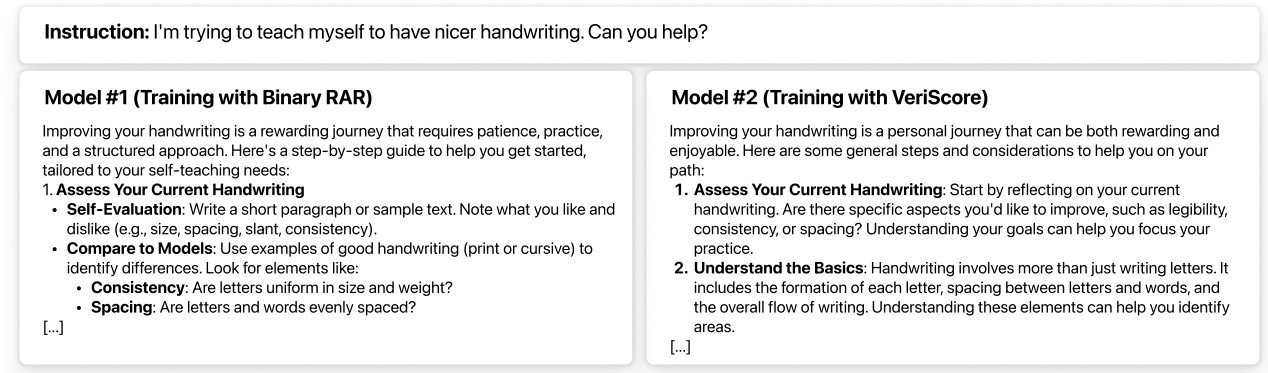

*Figure 7.* Comparing model outputs trained with Binary RAR and VeriScore. The Binary RAR model gives detailed, structured guidance, while the VeriScore model produces slightly high-level text.

**Binary RAR Reduces Hallucination While Preserving Detail.** Figure 8 compares outputs from Qwen3-8B before and after RL fine-tuning with Binary RAR. The base model generates incorrect information about Connecticut and Rhode

Island, whereas the fine-tuned model avoids these errors while adding relevant examples of states named after royalty. This demonstrates that RL fine-tuning with Binary RAR effectively reduces factual errors without sacrificing informative content—a crucial advantage over the alternatives explored above.

---

**Instruction:** How did US states get their names?

**Model #1 (Qwen3-8B)**

Many states were named after British or European royalty or their titles, often as a gesture of loyalty or to honor a monarch.
  · Virginia: Named after Queen Elizabeth I (the "Virgin Queen").
  · New York: Named after James Stuart, Duke of York (later King James II).
  · New Jersey: Named after John, Earl of Jersey (brother of the Duke of York).
  · Maryland: Named after Henrietta Maria, wife of King Charles I of England.
  · Louisiana: Named after King Louis XIV of France.
  · Connecticut: Named after the English county of Connecticut.
  · Rhode Island: Named after the Greek island of Rhodes.
  [...]

**Model #2 (Qwen3-8B RL finetuned with Binary RAR)**

[...]
2. European Colonial Influence
  · Louisiana: Named after King Louis XIV of France.
  · Maryland: Named after Henrietta Maria, wife of King Charles I of England.
  · New York: Named after the Duke of York (James, later King James II).
  · New Jersey: Named after the English earl of Jersey.
  · Pennsylvania: Named after William Penn, its founder.
  · Virginia: Named after Queen Elizabeth I (the "Virgin Queen").
  · West Virginia: Named after Virginia, as it was split from the original state.
  · Georgia: Named after King George II of England.
  · Washington: Named after George Washington.

---

*Figure 8.* Comparing Qwen3-8B before and after RL fine-tuning with Binary RAR. The fine-tuned model corrects factual errors and keeps relevant details, showing Binary RAR reduces hallucination without losing details.

## A.2. Analysis of the Verifier

To directly assess the verifier's error characteristics, we sampled 200 training prompts, generated responses with Qwen3-8B, and used GPT-5.4 to construct pseudo-ground-truth annotations of whether each response contradicted the retrieved evidence. Our main concern is false negatives (i.e., contradiction-free responses incorrectly assigned reward 0), because such errors can unnecessarily penalize correct responses and harm utility. Under this analysis, the false-negative rate is 16% for the Qwen3-32B verifier, compared with 30% for the Qwen3-8B verifier. This supports our choice of the stronger verifier and suggests that the observed gains depend in part on verifier quality.

## A.3. Analysis of the Retrieval

Our method does not require high-quality retrieved evidence. In an analysis of 200 randomly sampled training prompts, using responses from Qwen3-8B and GPT-5.4 web search to construct pseudo-ground-truth annotations, we found that when a response contains a factual error, the retrieval pipeline includes the relevant evidence in only 53% of cases. When evidence is missing, the verifier often cannot identify a contradiction, so different responses tend to receive the same reward and the update is close to neutral rather than systematically harmful.

## A.4. Additional Model Families

To assess whether Binary RAR generalizes to other model families, we apply it to Tulu3-8B, which is built on top of the Llama-3.1-8B checkpoint and already trained with a reward model. Table 5 shows that even in this strong starting point, Binary RAR consistently improves factual precision on Biography and WildHallu. Utility metrics remain nearly unchanged, confirming that the method adds factuality without disturbing the model's strengths. This result supports the broader conclusion that Binary RAR is compatible with advanced preference-tuned models and brings reliable gains without inducing undesirable behavior.

| Model | Biography (Hallucination Ratio↓) | WildHallu (Hallucination Ratio↓) | AlpacaEval (LC Win Rate↑) |
|---|---|---|---|
| Tulu3-8B (base) | 68.1 | 43.4 | 24.3 |
| + Binary RAR | 60.1 | 39.3 | 23.4 |

*Table 5.* RL with Binary RAR on Tulu3-8B.

## A.5. Preservation of Reasoning Length

We further analyze how training affects the model's reasoning traces. Using AIME25, we sample four outputs per problem and compare accuracy, reasoning length, and final response length (excluding reasoning) before and after RL training. Accuracy is largely preserved (slightly improved on Qwen3-8B and slightly reduced on Qwen3-4B), and response length is similar across both models. Reasoning length decreases modestly, which we attribute to the non-stop sequence penalty used during RL training.

| Model | AIME25 (Accuracy↑) | Reasoning Length (tokens) | Response Length (tokens) |
|---|---|---|---|
| Qwen3-8B | 64.1 | 14833 | 1130 |
| + Binary RAR | 68.3 | 13622 | 1079 |
| Qwen3-4B | 62.5 | 14618 | 1053 |
| + Binary RAR | 60.0 | 13473 | 1087 |

*Table 6.* Accuracy and reasoning/response lengths on AIME25.

# B. Evaluation Details

## B.1. Datasets

We assess hallucination in both long-form generation and short-form question answering using the following benchmarks:

- BIOGRAPHY (Min et al., 2023): A benchmark consisting of prompts that ask models to write biographies of specific individuals. We use `gpt-4.1` with a customized prompt to extract atomic claims, retrieve the top 10 evidence chunks (100 words each) for each claim, and use `gpt-4.1-mini` to verify whether each claim is supported by the retrieved evidence.

- WILDHALLUCINATION (Zhao et al., 2024): A dataset probing factual consistency across diverse real-world entities, including people, geography, and computing, with emphasis on rare entities.

- POPQA (Mallen et al., 2023): A short-form QA dataset covering entities of varying popularity. POPQA requires generating an entity (for example, a name, time, or occupation) without provided options, so we use `gpt-4.1` to judge whether the model response matches any ground-truth answer.

- GPQA (Rein et al., 2024): A multiple-choice QA dataset covering graduate-level biology, chemistry, and physics, where questions and answers are expert-authored. We mark an answer as correct if the model output includes the ground-truth option letter.

To measure whether factuality improvements cause regressions in other areas, we evaluate general capabilities using these benchmarks:

- ALPACAEVAL (Dubois et al., 2024): We use version 2 (v2) and report the length-controlled win rate metric to reduce length bias. The LM judge is `gpt-4.1`.

- ARENAHARD (Li et al., 2025): We use version 2.0 and report the style-controlled score. To ensure fair comparison, we add all baselines and our method to the official leaderboard and recompute the regression for style control.

- IFEVAL (Zhou et al., 2023): A benchmark of 500 prompts covering 25 types of verifiable instructions, designed to test instruction fidelity with objectively checkable outcomes.

- GSM8K (Cobbe et al., 2021): A dataset of grade-school math word problems requiring multi-step reasoning.

- MINERVA (Lewkowycz et al., 2022): A collection of 272 graduate-level quantitative reasoning problems in STEM fields such as physics and chemistry, requiring domain-specific expertise.

- HUMANEVAL (Chen et al., 2021): We use HumanEval+, an augmented version of HumanEval that adds additional test cases to improve robustness. Each problem includes multiple functional tests.

- MBPP (Austin et al., 2021): We use MBPP+, an augmented version of MBPP where each instance is equipped with more test cases.

### B.2. Abstention Detection

For PopQA, we use an LM judge to classify each output as correct, incorrect, or abstaining. Since PopQA uses short answers, abstentions appear in a very direct form. For GPQA, which is multiple choice, we explicitly prompt the model to output either one option or the exact phrase "I don't know." We manually inspected 50 outputs from each dataset and find that we were able to correctly identify abstentions in nearly all cases.

## C. Training Details

We perform continual RL fine-tuning on Qwen3-8B and Qwen3-4B, two reasoning LMs. GRPO serves as the main RL algorithm. We use Qwen3-32B as the verifier to compute binary RAR, prompting it to identify contradictions between model responses and retrieved documents. The learning rate is set to $1 \times 10^{-6}$, with KL coefficients of $1 \times 10^{-3}$ for Qwen3-8B and $3 \times 10^{-3}$ for Qwen3-4B. To compute binary RAR, we use BM25 retrieval with documents chunked into 512 tokens (using the Qwen3 tokenizer). For each response, we retrieve the top 8 chunks and verify the response with Qwen3-32B. We apply early stopping to prevent overtraining that could degrade utility. Specifically, training is stopped if a checkpoint exhibits more than a 10% drop on any utility benchmark. To compute VeriScore, we apply BM25 for retrieval, split documents into 256-token chunks (using the Qwen3 tokenizer), and retrieve the top 4 chunks per claim for verification. Both claim extraction and verification use Qwen3-32B.

**RL Fine-tuning.** We fine-tune models using reinforcement learning for up to four epochs, with a batch size of 16 unique prompts and 8 rollouts per prompt. Training typically runs for 2,000 steps, except for dense VeriScore rewards, where early stopping at 1,000 steps prevents degradation on utility benchmarks.

**SFT and DPO Baselines.** For supervised fine-tuning (SFT), one epoch provides the best balance between stability and performance. Direct preference optimization (DPO) is trained for four epochs with factuality-driven preference pairs.

**Retrieval and Pre-caching Strategy.** Both retrieval and verification are computationally intensive, and computing reward $r(x, y)$ can easily become the bottleneck of RL training. To improve efficiency, we adopt a pre-caching strategy. During dataset preparation, we pre-cache a set of relevant documents $\mathcal{DS}_{\text{cache}}(x)$ for each prompt $x$ in the training set $\mathcal{D}$. At training time, we retrieve $C(x, y)$ from this cached subset rather than from the full datastore $\mathcal{DS}$. To build $\mathcal{DS}_{\text{cache}}(x)$, we query the Google Search API using the ground-truth response to retrieve up to 10 potentially relevant web pages, which we crawl and parse using a rule-based Python pipeline. Instances with fewer than three retrieved documents are discarded, as sparse evidence is often insufficient for reliable verification. Each selected training prompt is thus paired with a compact, verified document set $\mathcal{DS}_{\text{cache}}(x)$ indexed by a BM25 retriever. Using a pre-caching strategy, we may not capture all possible information during training, but including relevant documents for each instance ensures a high chance that retrieved evidence will reveal contradictions in incorrect model outputs.

**Verification without Claim Decomposition.** Instead of extracting and verifying individual claims (as done in VeriScore), we detect contradictions by comparing the entire response with the retrieved documents in a single LM forward pass. This avoids repeated document processing and greatly reduces computation compared to concurrent work using VeriScore as a factuality reward (Chen et al., 2025). Binary RAR achieves a $2\times$–$4\times$ throughput improvement depending on response length, using four replicas of Qwen3-32B as the verifier on a cluster of 8 NVIDIA H100 GPUs.

## D. Reward Implementation

**Data Curation.** We curate instruction–response pairs from the WILDCHAT dataset (Zhao et al., 2024) and filter examples with verifiable factual content using `gpt-4.1`. For each prompt, we pre-cache retrieved documents using the Google Search API, retaining up to 10 relevant web pages. Instances with fewer than three reliable documents are discarded to ensure verification quality. The final curation yields diverse, factual prompts spanning entities, events, and scientific concepts.

**Reward Computation.** Each instruction–response pair $(x, y)$ is scored by comparing the response against retrieved documents using a verifier LM. The reward is binary:

$$r(x, y) = \begin{cases} 1, & \text{if no contradictions are found between } (x, y) \text{ and retrieved evidence,} \\ 0, & \text{otherwise.} \end{cases}$$

We use Qwen3-32B as the verifier with BM25 retrieval over 512-token chunks (Qwen3 tokenizer). Eight documents are retrieved per instance. This simple binary signal avoids partial credit and reduces noise from verifier bias. For efficiency, each prompt's retrieved set is pre-cached to reduce online retrieval overhead.

**Reward Prompts.** Figures 9 and 10 show the full prompts used for binary and rating-based retrieval-augmented rewards. These templates define the scoring logic, consistency rules, and JSON output structure for the verifier.

## E. More Discussion on Related Work

**Measuring hallucinations in LM outputs** Despite their impressive capabilities across diverse tasks, LMs are prone to hallucination, producing incorrect statements with unwarranted confidence (Mallen et al., 2023). The most widely adopted taxonomy distinguishes between two primary types of hallucination based on their relationship to provided prompts (Ji et al., 2023a; Huang et al., 2025; Bang et al., 2025). *Intrinsic hallucination* is defined as output that is inconsistent with the user's prompt or the provided input context. In this paper, we focus on *extrinsic hallucination*, which refers to generated output that cannot be verified from the training data. Measuring extrinsic hallucinations in long-form generation is particularly challenging due to its open-ended nature (Qi et al., 2025). Several distinct approaches have been proposed to automatically identify hallucinated content, including NLI-based methods (Gao et al., 2023; Min et al., 2023; Song et al., 2024), QA-based methods (Tian et al., 2024), uncertainty estimation (Farquhar et al., 2024; Orgad et al., 2025), and LLM-as-a-Judge (Li et al., 2024b). Following previous work, we adopt the approach of verifying atomic claims in the output as our evaluation method for long-form generation, which was first proposed in (Min et al., 2023). Specifically, we decompose a response into atomic, verifiable claims and then check each claim against related documents.

## F. Proof of Theorem 1

We restate the theorem in a fully formal form and then prove it by (i) recalling the standard closed-form optimizer for a KL-regularized expected-reward objective and (ii) substituting the binary reward.

**Theorem 1** (restate). *Fix $x$ and let $\mathcal{Y}_x = \{y : \pi_{\text{ref}}(y \mid x) > 0\}$ be the support of the reference policy. For $\beta > 0$, define the optimization problem over distributions $\pi(\cdot \mid x)$ supported on $\mathcal{Y}_x$:*

$$\max_{\pi(\cdot \mid x)} \sum_{y \in \mathcal{Y}_x} \pi(y \mid x)\, r(x, y) - \beta \sum_{y \in \mathcal{Y}_x} \pi(y \mid x) \log \frac{\pi(y \mid x)}{\pi_{\text{ref}}(y \mid x)} \quad s.t. \quad \sum_{y \in \mathcal{Y}_x} \pi(y \mid x) = 1,\ \pi(y \mid x) \geq 0. \tag{8}$$

*Assume $r(x, y) \in \{0, 1\}$ for all $y \in \mathcal{Y}_x$. Then the problem has a unique maximizer $\pi^\star(\cdot \mid x)$, and for all $y \in \mathcal{Y}_x$,*

$$\pi^\star(y \mid x) \ \propto\ \pi_{\text{ref}}(y \mid x) \times \begin{cases} 1 & \text{if } r(x, y) = 1, \\ \exp(-1/\beta) & \text{if } r(x, y) = 0. \end{cases} \tag{9}$$

**Recall: KL-regularized reward has a Gibbs-form optimizer.** A standard result (see, e.g., Rafailov et al. 2023) is that for any bounded reward $r(x, \cdot)$, the unique maximizer of Equation 8 is

$$\pi^\star(y \mid x) = \frac{\pi_{\text{ref}}(y \mid x) \exp(r(x, y)/\beta)}{\sum_{y' \in \mathcal{Y}_x} \pi_{\text{ref}}(y' \mid x) \exp(r(x, y')/\beta)}. \tag{10}$$

For completeness, we prove Equation 10 directly below, and then derive Equation 9 as a corollary for $r \in \{0, 1\}$.

*Proof.* Fix $x$ and abbreviate $\pi(y) = \pi(y \mid x)$, $\pi_{\text{ref}}(y) = \pi_{\text{ref}}(y \mid x)$, and $r(y) = r(x, y)$ on the set $\mathcal{Y}_x$. Expanding the objective in Equation 8, we maximize

$$J(\pi) = \sum_{y \in \mathcal{Y}_x} \pi(y)\, r(y) - \beta \sum_{y \in \mathcal{Y}_x} \pi(y) \log \frac{\pi(y)}{\pi_{\text{ref}}(y)} \quad \text{over} \quad \Delta(\mathcal{Y}_x) = \Big\{ \pi : \sum_y \pi(y) = 1,\ \pi(y) \geq 0 \Big\}. \tag{11}$$

Since $\beta > 0$ and $\pi_{\text{ref}}(y) > 0$ on $\mathcal{Y}_x$, the term $-\sum_y \pi(y) \log \pi(y)$ is strictly concave on $\Delta(\mathcal{Y}_x)$, and the remaining terms are linear in $\pi$. Therefore $J(\pi)$ is strictly concave on $\Delta(\mathcal{Y}_x)$, so it has a unique maximizer. It suffices to find a stationary point under the simplex constraint.

Introduce a Lagrange multiplier $\lambda$ for $\sum_y \pi(y) = 1$ and define

$$\mathcal{L}(\pi, \lambda) = \sum_y \pi(y)\, r(y) - \beta \sum_y \pi(y) \log \frac{\pi(y)}{\pi_{\text{ref}}(y)} + \lambda \Big( \sum_y \pi(y) - 1 \Big), \tag{12}$$

where all sums are over $y \in \mathcal{Y}_x$. For any $y$ with $\pi(y) > 0$, differentiating and setting to zero gives

$$0 = \frac{\partial \mathcal{L}}{\partial \pi(y)} = r(y) - \beta \Big( \log \frac{\pi(y)}{\pi_{\text{ref}}(y)} + 1 \Big) + \lambda. \tag{13}$$

Rearranging,

$$\pi(y) = \pi_{\text{ref}}(y) \exp\Big( \frac{r(y)}{\beta} \Big) \exp\Big( \frac{\lambda - \beta}{\beta} \Big). \tag{14}$$

Let $c = \exp((\lambda - \beta)/\beta)$ (a constant independent of $y$). Enforcing $\sum_y \pi(y) = 1$ yields

$$1 = \sum_y \pi(y) = c \sum_y \pi_{\text{ref}}(y) \exp(r(y)/\beta),$$

so

$$c = \Big( \sum_{y'} \pi_{\text{ref}}(y') \exp(r(y')/\beta) \Big)^{-1}.$$

Substituting back into Equation 14 gives exactly Equation 10. Uniqueness follows from strict concavity.

Finally, apply $r(y) \in \{0, 1\}$. Then $\exp(r(y)/\beta)$ equals $\exp(1/\beta)$ when $r(y) = 1$ and equals 1 when $r(y) = 0$. Factoring out $\exp(1/\beta)$ from the numerator and denominator of Equation 10 gives the equivalent reweighting

$$\pi^\star(y) \propto \pi_{\text{ref}}(y) \times \begin{cases} 1 & r(y) = 1, \\ \exp(-1/\beta) & r(y) = 0, \end{cases}$$

which is Equation 9. $\qquad\square$

## Binary Retrieval-augmented Reward

You are a professional fact-checker tasked with evaluating a language model's response for accuracy. Your job is to systematically compare the response against the provided web search results to identify any factual errors or contradictions. A response deserves a full score if every claim it makes is factually correct, even if it doesn't include every available detail. Omitting information is not a factual error. If a response contains anything that contradicts the world knowledge shown in the search results, it should be considered an error and get a low score.

## EVALUATION PROCESS:
1. Read the search results thoroughly to understand the factual baseline
2. Examine each factual claim in the language model's response
3. Cross-reference each claim against the search results
4. Classify each discrepancy according to the guidelines below

## CONSIDER CORRECT (No Factual Error):
- Paraphrasing: Same facts expressed in different words
- Reasonable inferences: Logical conclusions drawn from search results
- Partial information: Incomplete but accurate subsets of available information
- Contextual additions: Background information that doesn't contradict search results
- Minor formatting differences: Different ways of presenting same data

## CONSIDER INCORRECT (Factual Error):
- Direct contradictions: Response states opposite of what search results show
- Numerical errors: Wrong dates, statistics, quantities, percentages
- Categorical errors: Wrong classifications, locations, names, titles
- Causal errors: Incorrect cause-and-effect relationships
- Timeline errors: Events placed in wrong chronological order
- Attribution errors: Wrong sources, authors, or speakers cited

## SCORING RULES:
Your evaluation will result in a binary score: 0 or 1.
- SCORE 1 (No Contradiction): Assign this score if:
1. The response is fully supported by the document.
2. The response contains information that is NOT in the document, but DOES NOT contradict it.
- SCORE 0 (Contradiction): Assign this score ONLY if you find a clear, factual contradiction between the response and the supporting document. A contradiction occurs when the response states the opposite of what the document says (e.g., wrong dates, names, events, or outcomes).

## OUTPUT FORMAT:
You must respond ONLY in a valid JSON format with exactly these two fields:
- "REASONING": A brief explanation for your score.
- For SCORE 0, specify the contradiction (e.g., "The response states the event was in 2022, but the document says it was in 2023.").
- For SCORE 1, simply state "No contradiction found."
- "SCORE": An integer, either 0 or 1.

>>> Begin of search results <<<
{passages_text}
<<< End of search results >>>

>>> Begin of the prompt <<<
{prompt_text}
<<< End of the prompt >>>

>>> Begin of the response <<<
{response_text}
<<< End of the response >>>

Respond in JSON format. {{"REASONING": "[...]", "SCORE": "<your-score>"}}

*Figure 9.* Prompt for binary fact-checking.

---

**Rating Retrieval-augmented Reward**

You are a professional fact-checker tasked with evaluating a language model's response for accuracy. Your job is to systematically compare the response against the provided web search results to identify any factual errors or contradictions. A response deserves a full score if every claim it makes is factually correct, even if it doesn't include every available detail. Omitting information is not a factual error. If a response contains anything that contradicts the world knowledge shown in the search results, it should be considered an error and get a low score.

## EVALUATION PROCESS:
1. Read the search results thoroughly to understand the factual baseline
2. Examine each factual claim in the language model's response
3. Cross-reference each claim against the search results
4. Classify each discrepancy according to the guidelines below

## CONSIDER CORRECT (No Factual Error):
- Paraphrasing: Same facts expressed in different words
- Reasonable inferences: Logical conclusions drawn from search results
- Partial information: Incomplete but accurate subsets of available information
- Contextual additions: Background information that doesn't contradict search results
- Minor formatting differences: Different ways of presenting same data

## CONSIDER INCORRECT (Factual Error):
- Direct contradictions: Response states opposite of what search results show
- Numerical errors: Wrong dates, statistics, quantities, percentages
- Categorical errors: Wrong classifications, locations, names, titles
- Causal errors: Incorrect cause-and-effect relationships
- Timeline errors: Events placed in wrong chronological order
- Attribution errors: Wrong sources, authors, or speakers cited

## CONFIDENCE SCORING GUIDE:
- 0-2: Very confident there is a factual error (multiple clear contradictions)
- 3-4: Moderately confident there is a factual error (one clear contradiction)
- 5: Uncertain (ambiguous evidence or unclear from search results)
- 6-7: Moderately confident there is no factual error (mostly accurate with minor concerns)
- 8-10: Very confident there is no factual error (all stated facts are accurate, regardless of completeness)

## OUTPUT FORMAT REQUIREMENTS:
Respond ONLY in valid JSON format with exactly these two fields:
- "REASONING": A concise explanation of your assessment (1-2 sentences max, e.g., "the response states ... but the search results show ... so there is a factual error" or "no factual error found")
- "SCORE": An integer from 0-10 representing your confidence level

>>> Begin of search results <<<
{passages_text}
<<< End of search results >>>

>>> Begin of the prompt <<<
{prompt_text}
<<< End of the prompt >>>

>>> Begin of the response <<<
{response_text}
<<< End of the response >>>

Respond in JSON format. {{"REASONING": "[...]", "SCORE": "<your-score>"}}

---

*Figure 10.* Prompt for rating-based fact-checking.

---

## Claim Extraction for VeriScore Training / FactScore Evaluation

Extract as many fine-grained, atomic, and verifiable factual claims as possible from the response. Each claim should be a single piece of information that could be looked up in a database, official documentation, reputable forum, or reliable source such as Wikipedia or scientific literature.

**Guidelines for atomic claims:**
- Split a sentence that joins different facts using "and," "or," or by listing into multiple claims.
- If a claim could be split into multiple smaller, independent statements, do so.
- Replace pronouns (e.g., "he", "she", "it", "they") with the full entity name explicitly stated in the response. If the entity name is not explicitly mentioned, leave the pronoun unchanged.
- Extract claims EXACTLY as stated, even if the information appears incorrect or false.

**Include as claims:**
- Statements about the existence, property, function, or relationship of entities, organizations, concepts, or technologies.
- Claims about names, definitions, features, purposes, or histories.
- Statements about what something does, who runs it, what it is used for, or what it affects.
- For hedged language ("may be," "might be," "could be"), extract the factual association, typical usage, or commonly reported function as long as the claim is traceable to community consensus, documentation, or reputable user reports.
- If a quotation is present, extract it verbatim with the source if given.
- Claims must stand alone, using names or clear descriptions, not pronouns.

**Do not include as claims:**
- Personal opinions, suggestions, advice, instructions, or experiences.
- Pure speculation or possibilities that are not reported in any documentation or user discussions.
- Claims from code blocks or pure math derivations.

Extract claims only from the response section, not from the prompt or question. If the response does not contain any verifiable factual claims, output an empty list.

Output a JSON list of strings. Each string should be a single atomic factual claim from the response, clearly stated and verifiable.

>>> Begin of prompt <<<
{prompt_text}
<<< End of prompt >>>

>>> Begin of response <<<
{response_text}
<<< End of response >>>

Facts (as a JSON list of strings):

*Figure 11.* Prompt for atomic claim extraction.

---

**Claim Verification for VeriScore Training / FactScore Evaluation**

You need to judge whether a claim is supported or contradicted by Google search results, or whether there is no enough information to make the judgement. When doing the task, take into consideration whether the link of the search result is of a trustworthy source.

Below are the definitions of the three categories:

Supported: A claim is supported by the search results if everything in the claim is supported and nothing is contradicted by the search results. There can be some search results that are not fully related to the claim.
Contradicted: A claim is contradicted by the search results if something in the claim is contradicted by some search results. There should be no search result that supports the same part.
Inconclusive: A claim is inconclusive based on the search results if:
- a part of a claim cannot be verified by the search results,
- a part of a claim is supported and contradicted by different pieces of evidence,
- the entity/person mentioned in the claim has no clear referent (e.g., "the approach", "Emily", "a book").

>>> Begin of search results <<<
{passages_text}
<<< End of search results >>>

Claim: {claim_text}
Task: Given the search results above, is the claim supported, contradicted, or inconclusive? Your answer should be either "supported", "contradicted", or "inconclusive" without explanation and comments.

Your decision:

*Figure 12.* Prompt for claim verification.

