# OpenReview forum: "Train for Truth, Keep the Skills: Binary Retrieval-Augmented Reward Mitigates Hallucinations"
_ICML.cc/2026/Conference — ICML 2026 spotlight_

### Official Review · Reviewer_rMaN · 2026-03-06

**Soundness:** 4
**Presentation:** 4
**Significance:** 2
**Originality:** 2
**Overall Recommendation:** 4
**Confidence:** 3

**Summary:**

The paper introduces Binary Retrieval-Augmented Reward, that assigns a score of 0/1 to reduce hallucinations while preserving general capabilities. To compute this binary reward, the authors build a corpus of evidence from internet and judge the responses against this corpus using an LLM during post training. The paper further claims that binary rewards are more robust than continuous alternatives and LLM-as-a-judge methods at avoiding reward hacking and preserving utility.

**Compliance With Llm Reviewing Policy:**

Affirmed.

**Final Justification:**

Authors have addressed my concerns around hallucination reduction and performance gains. They show on AIME ~8% reduction in response length for thinking tokens while score remains comparable with baseline. I have updated my score from 3->4.

**Key Questions For Authors:**

Please refer Strengths And Weaknesses section

**Limitations:**

yes

**Strengths And Weaknesses:**

### Strengths
---
- Paper is well written and method is easy to follow.
- Authors provide detailed ablations and mathematical grounding to their approach.

### Weakness
---
- Binary RAR reduces the hallucination rate but the model's performance on general benchmarks remains similar to the base model. This suggests that the presence of factually incorrect information does not adversely affect the base model's performance on downstream tasks. It would be beneficial to study this interplay. It also raises the question whether majority of gain in hallucination reduction by Binary RAR comes from the abstention behaviour and model not learning to output factually correct information?
- Binary RAR has strict reliance on KL penalty to prevent performance degradation. This limits the utility of Binary RAR in post training pipelines like DAPO which suggest removing KL penalty. Further, the paper do not study the combination of correctness reward and Binary RAR (or VeriScore) which could be beneficial for preserving model utility.
- I would also be interested to see how Binary RAR trained model performs in RAG setup and if we observe any abstention behaviour in that setting.
- How do authors make sure that 8 retrieved evidence chunks has all the relevant information for the <Prompt, Response> pair?
- Minor: Incorrect model name on line 252 Qwen3-7B -> Qwen3-8B

---

> ### Author Rebuttal · Authors · 2026-03-31
>
> We sincerely thank Reviewer rMaN for the thoughtful review. We appreciate the reviewer’s positive comments on the **clarity of the paper**, the **detailed ablations**, and the **mathematical grounding of the approach**. We address the main points below.
>
> - (W1) **Interplay between hallucination reduction and utility.** Reviewer rMaN asks whether factual errors do not matter for downstream tasks since it remains the same, and whether the gains mainly come from abstention rather than improved factual correctness.
>     - **The main goal of this work is to reduce hallucinations without hurting other capabilities that are largely orthogonal to hallucination.** The utility benchmarks are designed to measure this tradeoff. Most of them either do not depend heavily on world knowledge (e.g., math and coding), or they do not penalize hallucinated answers (e.g., GPQA). Our results show that Binary RAR preserves utility well, whereas other baselines can significantly hurt it. For example, on AlpacaEval, Binary RAR achieves 53.9, while VeriScore drops to 42.2.
>     - **In long-form generation in particular, our goal is not to abstain, but to preserve useful content while removing hallucinated content.** On the Biography dataset, the average number of claims decreases from 30.0 to 13.6 after training (Figure 3), while the number of correct claims remains nearly unchanged, from 8.8 to 8.6. This indicates that Binary RAR mainly removes hallucinated statements rather than improving hallucination rates simply by abstaining or becoming uninformative.
> - (W2) **Reliance on KL regularization.** Reviewer rMaN asks whether Binary RAR depends too strongly on KL regularization to avoid performance degradation, which may limit its compatibility with post-training pipelines such as DAPO.
>     - Our setting is fully compatible with DAPO. DAPO is designed to improve reasoning ability starting from an SFT model, whereas our setting starts from an already post-trained model and applies an additional RL stage to reduce hallucinations while preserving existing general capabilities. **Our method can naturally be applied after DAPO.**
>     - **Binary RAR reward itself is agnostic to the underlying RL optimizer** and can be combined with different RL algorithms, including PPO- or GRPO-style training.
>     - The role of KL is also different in the two settings. In our setting, KL regularization is useful because it limits unnecessary drift from a strong base model during hallucination reduction training. In contrast, DAPO uses low or zero KL to encourage exploration when improving reasoning from an earlier-stage model.
> - (W2) **Combining Binary RAR with correctness rewards.** Reviewer rMaN asks whether adding a correctness-style reward to Binary RAR or VeriScore could further improve utility preservation.
>     - In our setting, training uses chat-style prompts from WildChat, which do not provide reliable ground-truth correctness labels. Binary RAR already achieves a strong utility-hallucination tradeoff without requiring an additional correctness or utility reward. By contrast, methods such as VeriScore often benefit from extra LM-judge-style signals to better preserve utility.
> - (W3) **Binary RAR in a RAG setting.** Reviewer rMaN asks how a Binary RAR-trained model would behave in a RAG setting, including whether it would exhibit abstention behavior.
>     - Our work focuses on extrinsic hallucination, where claims must be checked against external evidence not given in the prompt. A RAG setting is more closely related to intrinsic hallucination: the model is given the relevant document set, and the key question is whether it answers consistently with that provided context. Because these settings differ, we do not study RAG behavior here, but view it as an interesting direction for future work.
> - (W4) **Retrieval coverage**. Reviewer rMaN asks whether the top-8 retrieved chunks are sufficient to cover all information needed to verify a given prompt-response pair.
>     - Our method does not require high retrieval coverage for every example. In an additional analysis of 200 randomly sampled training prompts, using responses from Qwen3-8B and GPT-5.4 web search to construct pseudo-ground-truth annotations, we found that **when a response contains a factual error, the retrieval pipeline includes the relevant evidence in only 53% of cases**. This shows that retrieval is often incomplete in practice.
>     - The method remains effective because **the reward is contradiction-based**. If the retrieved corpus lacks relevant evidence, different responses tend to receive the same reward, so the update is close to neutral and the model behavior remains largely unchanged.
> - (W5) Minor: Incorrect model name on line 252 Qwen3-7B -> Qwen3-8B
>     - Thank you for catching this. We will correct the typo in the revision.

---

> > ### Author Rebuttal · Reviewer_rMaN · 2026-04-02
> >
> > Thank you for the response. I remain unconvinced on W1: the hallucination reduction primarily comes from generating less content rather than producing more factually correct outputs on downstream tasks. Can the authors demonstrate a benchmark where hallucination reduction from Binary RAR translates to actual performance improvement? For example, authors could evaluate their model and baselines on AIME 24/25 (30 problems) to solidify the utility preservation claim, and show a response length vs. accuracy tradeoff similar to Figure 3 (right)?
> >
> > --- UPDATE ---
> >
> > Authors have addressed my comments AIME shows ~8% reduction in response length while score remains comparable with baseline. I have updated my score from 3->4.

---

> > > ### Author Response · Authors · 2026-04-03
> > >
> > > We thank the reviewer for the follow-up and address the two main concerns below.
> > >
> > > Why is Binary RAR not designed to encourage models to generate more factually correct information?
> > >
> > > - **The case study in Fig. 8 might better illustrate the goal of our work.** When prompted with "How did US states get their names?" (an example from AlpacaEval), in the section of European colonial influence, Qwen3-8B correctly describes the origins of five states (Virginia, New York, New Jersey, Maryland, Louisiana) but makes incorrect claims about Connecticut and Rhode Island. After training on Binary RAR (Fig. 8, right), the model retains all correct statements while removing incorrect ones.
> > > - **Our method is based on the assumption that the fully trained base model (from which Binary RAR training begins) has already been optimized for accuracy across tasks.** For knowledge-intensive tasks in particular, it is common practice to optimize either for accuracy [1] in short-form QA or for coverage of rubric items [2] in long-form generation.
> > > - In other words, **the fully trained base model is already capable of generating most of the relevant knowledge it has acquired.** **The goal of Binary RAR is therefore to preserve correct and relevant content while removing hallucinated content.** This design is also motivated by recent findings showing that reasoning models and heavily post-trained models optimized for accuracy can hallucinate more [3].
> > >
> > > How does training with Binary RAR translate to downstream tasks?
> > >
> > > - We evaluate on three types of tasks. For **knowledge-intensive long-form generation** (e.g., Biography, AlpacaEval), Binary RAR reduces incorrect claims (i.e., hallucinations; Tab. 2), preserves the number of correct claims (Fig. 3), and maintains LM-judge scores (Tab. 3).
> > > - For **knowledge-intensive short-form QA** (e.g., PopQA, GPQA), Binary RAR preserves correct responses and converts most incorrect responses into abstentions (Fig. 4).
> > > - For **non-factual tasks, including reasoning and coding**, accuracy remains unchanged (Tab. 3). We attribute this to the fact that outputs change only minimally on non-factual tasks. Our training data includes only prompts requiring external factual knowledge, as discussed in Sec. 4.2, and KL regularization helps the model preserve its original behavior.
> > >
> > > To further show that outputs on non-factual tasks change only minimally, we include an analysis on AIME25, as suggested by the reviewer. For each problem, we sampled four responses before and after RL training and compared both accuracy and average output length. We find that accuracy and response length remain similar overall. We do observe a slight reduction in reasoning length, which may be related to the penalty on non-stop sequences used during RL training.
> > >
> > > | AIME25 | Accuracy | Reasoning Length (tokens) | Response Length (tokens) |
> > > | --- | --- | --- | --- |
> > > | Qwen3-8B | 64.1 | 14833 | 1130 |
> > > | + Binary RAR | 68.3 | 13622 | 1079 |
> > > | Qwen3-4B | 62.5 | 14618 | 1053 |
> > > | + Binary RAR | 60.0 | 13473 | 1087 |
> > >
> > > - [1] Lambert et al. (2024), Tulu 3: Pushing Frontiers in Open Language Model Post-Training
> > > - [2] Gunjal et al. (2025), Rubrics as Rewards: Reinforcement Learning Beyond Verifiable Domains
> > > - [3] Kalai et al. (2025), Why Language Models Hallucinate

---

### Official Review · Reviewer_USKd · 2026-03-06

**Soundness:** 2
**Presentation:** 3
**Significance:** 2
**Originality:** 2
**Overall Recommendation:** 4
**Confidence:** 4

**Summary:**

This paper proposes Binary RAR, an online RL approach that assigns a binary reward based on whether a generated response contradicts retrieved evidence.
The method aims to reduce hallucinations while preserving general capabilities by penalizing responses containing factual contradictions. Experiments on Qwen3 models show reduced hallucination rates on long-form and short-form QA benchmarks while maintaining performance on several capability benchmarks.

**Compliance With Llm Reviewing Policy:**

Affirmed.

**Final Justification:**

The rebuttal provides helpful clarifications and some additional analyses. While some concerns remain regarding the novelty of factuality rewards and potential reliance on the verifier, the overall approach appears reasonably explained.
I consider the paper borderline but slightly leaning positive.

**Key Questions For Authors:**

Please refer to weakness.

**Limitations:**

yes

**Strengths And Weaknesses:**

# Strength
The paper addresses an important problem of hallucination in LLM post-training. The proposed reward design is simple but practical, and appears empirically effective across short-form and long-form answers.

# Weakness

- **Limited novelty.**

    The core idea, penalizing incorrect responses using a simple reward signal, appears conceptually close to prior approaches such as RLKF and RL with factuality rewards. The paper does not clearly articulate a deeper conceptual or theoretical distinction beyond discretizing the reward signal. In addition, the discussion around KL regularization does not constitute a novel contribution, as KL constraints are already a standard and important component of existing RL-based post-training frameworks (e.g., GRPO and RLHF). As a result, the overall methodological novelty appears limited.

- **Baseline comparisons may not be fully fair or sufficient.**

    The comparison with continuous-reward baselines (e.g., VeriScore) does not clearly ensure equivalent KL constraints or comparable reward scaling. Without controlling for these factors, it is difficult to determine whether the reported improvements are due to the binary reward formulation itself or differences in training dynamics and optimization stability.

- **Dependence on an LM verifier introduces potential bias.**

    The reward relies on contradiction judgments from a large verifier model. However, the reliability and error characteristics of this verifier are not analyzed (unless I missed it). This raises concerns that the trained policy may partially overfit to the verifier’s biases rather than genuinely improving factual correctness.

---

> ### Author Rebuttal · Authors · 2026-03-31
>
> We sincerely thank Reviewer USKd for the thoughtful review. We appreciate the reviewer’s recognition that the paper addresses **an important problem** and that **the proposed method is simple, practical, and empirically effective**. We address the main concerns below.
>
> - (W1) **Limited novelty.** Reviewer USKd argues that our method is conceptually close to prior RL approaches (e.g., RLKF) that use factuality rewards, and that KL regularization is standard in RL post-training.
>     - Our contribution is not simply “a factuality reward” or “adding KL.” Prior work like RLKF and TruthRL focus on maximizing accuracy and encouraging abstention on certain benchmarks with ground-truth labels with targeted RL training. Whereas **we study a harder setting: continual RL training from trained models to reduce hallucinations for long-form generation without hurting other capabilities**. We found existing factuality RL recipes often significantly hurt the utility.
>     - Therefore, we introduced a new method that can better balance this trade-off. **We empirically find that two design choices are critical**: verifying contradictions rather than attribution, and scoring the whole response rather than decomposed claims. These choices make the reward substantially more robust to retrieval noise and claim-decomposition errors.
>     - **We theoretically show that the optimal policy preserves the relative distribution among contradiction-free responses** and downweights contradiction-containing responses by a constant determined by the KL coefficient. Binary RAR does not push the model toward vacuous answers and it specifically suppresses contradictory ones.
> - (W2) **Fair baseline comparisons.** Reviewer USKd asks whether the baseline comparisons are fair, and whether the gains over continuous rewards could instead come from differences in KL constraints, or reward scale.
>     - We provide fair comparisons in our analysis. **Figure 5** presents training dynamics comparing Binary RAR and VeriScore under matched training conditions, while varying the KL coefficient and training steps. Under these comparable optimization settings, Binary RAR lies on the Pareto frontier of the utility-hallucination tradeoff. This suggests that the observed gains are not simply due to more favorable optimization or weaker regularization, but instead reflect an advantage of the reward design itself.
>     - Reward scaling also does not explain the difference under GRPO. As shown in Equation 3, the advantage term is normalized by its standard deviation. As a result, multiplying all rewards by a constant leaves the normalized advantage unchanged, and therefore does not materially change the training dynamics.
>
> Table: Fair baseline comparisons (Figure 5)
> | Reward | KL Coefficient | Steps | Biography (Hallu. Ratio,↓) | AlpacaEval (LC Win Rate,↑) |
> | --- | --- | --- | --- | --- |
> | VeriScore | 0.001 | 1000 | 44.4 | 11.6 |
> | Binary RAR | 0.001 | 1000 | 58.8 | 53.7 |
> | Binary RAR | 0.001 | 2000 | 45.8 | 53.9 |
>
> - (W3) **LM verifier reliability.** Reviewer USKd raises a concern that the reliability and error characteristics of the verifier are not sufficiently analyzed, and that the model may overfit to verifier bias rather than genuine factual improvement.
>     - To directly assess the verifier’s error characteristics, we additionally sampled 200 training prompts, generated responses with Qwen3-8B, and used GPT-5.4 to construct pseudo-ground-truth annotations of whether each response contradicted the retrieved evidence. Our main concern is false negatives (i.e., contradiction-free responses incorrectly assigned reward 0), because such errors can unnecessarily penalize correct responses and harm utility. Under this analysis, the false-negative rate is 16% for the Qwen3-32B verifier, compared with 30% for the Qwen3-8B verifier. This supports our choice of the stronger verifier and suggests that the observed gains depend in part on verifier quality. We will add this analysis in the revision.
>     - We chose Qwen3-32B because it is an open-weight model, which improves reproducibility, and it was among the strongest models at this scale at the time of our experiments. We agree that improving verifier reliability is an important direction for future work, and stronger verifiers could likely further improve this framework.

---

> > ### Author Rebuttal · Reviewer_USKd · 2026-04-02
> >
> > Thanks for the authors’ response. I still have the following concern:
> >
> > Regarding the novelty, I would like to double-check (or ask for clarification). My understanding is that your main novelty lies in explicitly retrieving evidence and comparing it with the generated response during training to detect contradictions and then compute the reward.
> >
> > However, for the continual RL training aspect, it seems that your setup is not fundamentally different from standard RL with KL regularization. The “continual” property is mainly enforced through the KL constraint, which is a commonly used design in RL-based post-training. Therefore, it is unclear whether this part constitutes a novel contribution.
> >
> > If the method itself is not intended to change, I think it would be better to reduce the emphasis on KL in the paper.

---

> > > ### Author Response · Authors · 2026-04-03
> > >
> > > We thank the reviewer for the follow-up and the clear articulation of the concern.
> > >
> > > - **Problem context.** This work studies the hallucination-utility tradeoff in continual post-training, with the goal of reducing hallucinations in long-form generation without degrading general capabilities. This setting differs from prior work such as RLKF, which focuses on improving factuality in short-form QA starting from base models, and therefore introduces different challenges in reward design.
> > >
> > > - **On the role of KL regularization.** We agree that the RL algorithm and KL regularization are standard components of RL post-training, and we do not intend to suggest otherwise. **Our contribution regarding KL is an empirical and theoretical finding**: when paired with our binary reward, KL regularization alone is sufficient to preserve utility, without requiring explicit auxiliary rewards such as an LM judge or a detail-oriented reward. This contrasts with prior work such as FLAME and "Reason for Factuality", which relies on auxiliary reward signals to mitigate degradation. We will revise the paper to make this distinction clearer.
> > >
> > > - **Scope of novelty.** The reviewer asks whether the main contribution is "explicitly retrieving evidence and comparing it with the generated response to detect contradictions." This is one component, but the contribution is broader. We show that several design choices, namely contradiction-only verification, response-level rather than claim-level scoring, and binarization, are each important, and that their combination is what places Binary RAR on the Pareto frontier where the alternatives fail. Section 6.2 and Figure 4 demonstrate this directly: alternative reward formulations all degrade utility during training, making sustained RL optimization difficult.
> > >
> > > Table: Comparison of reward variants; see Figure 4 for detailed results.
> > > | Rewards | Biography (Hallucination Ratio,↓) | AlpacaEval (LC Win Rate,↑) |
> > > | --- | --- | --- |
> > > | VeriScore | 51.7 | 42.2 |
> > > | Binary VeriScore (binarized with a threshold of 0.5) | 61.3 | 48.2 |
> > > | Rating RAR (using a 0-10 verifier score) | 65.0 | 46.9 |
> > > | Binary RAR | 45.8 | 53.9 |
> > >
> > > - **Theoretical and practical support.** Our reward design is further supported by theoretical analysis and motivated by practical efficiency considerations. In Sec. 3.4, we formally show that the optimal policy reduces the probability of hallucinated responses by a constant determined by the KL coefficient while preserving the relative distribution among contradiction-free responses. **This property ensures that no spurious preference is introduced among contradiction-free responses**, thereby avoiding utility degradation. This also explains why auxiliary rewards are unnecessary under our formulation. On the practical side, reward computation (retrieval + verification) can dominate wall-clock time in long-form generation, often exceeding the cost of training itself. As discussed in Sec. 3.3, **our reward design improves throughput by 2-4x relative to VeriScore**, which is also important for scaling RL-based hallucination reduction.
> > >
> > > --- UPDATE ---
> > >
> > > We thank the reviewer for their review and thoughtful questions about our paper. We are happy to engage and address any additional questions. If our response has fully addressed your concerns, we would greatly appreciate it if you would consider reevaluating your assessment of the paper.

---

### Official Review · Reviewer_HaLC · 2026-03-13

**Soundness:** 3
**Presentation:** 3
**Significance:** 3
**Originality:** 2
**Overall Recommendation:** 5
**Confidence:** 4

**Summary:**

This paper proposes Binary Retrieval-Augmented Reward (Binary RAR), an online RL approach that reduces hallucinations while preserving a model’s general capabilities. Binary RAR assigns a reward of 1 if a response contains no factual contradictions with retrieved evidence and 0 otherwise, theoretically reducing error-containing responses without altering the distribution of correct ones. Experiments show substantial hallucination reductions and improved robustness compared to continuous factuality rewards (which are prone to reward hacking). The authors further show that their training paradigm does not negatively affect performance on tasks such as open-ended chat and reasoning, while preserving factual knowledge and eliciting the skill of calibrated abstention.

**Compliance With Llm Reviewing Policy:**

Affirmed.

**Final Justification:**

Authors have answered all my questions and run additional experiments to address my concerns. I am in favor of accepting the paper.

**Key Questions For Authors:**

**1. Binary vs Continuous Rewards**: Does the proposed method converge slower than approaches using continuous rewards? Binary feedback could make credit assignment more difficult compared to continuous signals that provide finer-grained supervision.

**2. Denser Credit Assignment**: If the verifier can determine whether a response contradicts the evidence, it should also have the capability to identify which specific parts of the response contain contradictions? If so, could this be used to perform more granular credit assignment (e.g., one could do token level advantages and increase negative weight on tokens which the verifier identifies as directly contradicting the evidence). Note that this would be different from the Rating-Based RAR that authors tried, as the objective here is not to get a continuous score for the entire response (which might be hackable as noted by authors), but instead to do credit assignment within a response. I understand that this might be hard to spin up during the rebuttal, so I am not expecting an additional experiment for this question, even a well-formed answer is okay. However, if authors are able to implement it, it would greatly strengthen the paper.

**3. Length Effects**: The reward design may encourage shorter outputs because fewer claims reduce the probability of contradictions. What prevents the model from producing empty or extremely short responses to maximize reward? Is the KL constraint the only mechanism preventing this? Can the authors show how the average number of claims evolve over training?

**4. Online DPO Baseline**: It would be interesting to see the performance of Online DPO on this task. Offline methods often underperform in RL settings, so the current DPO/SFT results are not surprising. Online DPO might both benefit from on-policy training and avoid the reward hacking issues of continuous reward signals such as VeriScore (because DPO would only use binary preference rewards).

**5. Retrieval Dependence**: How sensitive is the method to retrieval quality? For example, how often does the model contradict evidence that is not present in the retrieved document set?

**6. Figure Rendering**: Figure 1 appears to have rendering issues; the visualization behaves weirdly when zooming in and out of the paper. This might be on my end, but please double-check that this is not a general issue.

**Limitations:**

yes

**Strengths And Weaknesses:**

**Strengths**

**1. Clear Writing and Presentation**: The paper is well-written, easy to follow, and presents its ideas clearly. The motivation, method, and experimental setup are coherent and insightful to read.

**2. Extensive Empirical Results**: The authors evaluate their method across multiple settings, including open-ended chat, reasoning, and factual knowledge tasks, to verify that hallucination reduction does not degrade general capabilities. The ablations are well-designed and provide useful insights—for example, showing that seemingly reasonable reward designs (e.g., LLM-as-judge or VeriScore) are prone to hacking.

**Weaknesses**

**1. Reward Design and Credit Assignmen**t: Some design choices could be better justified. In particular, it is unclear why denser credit assignment cannot be implemented within the assumptions of the paper. Since the verifier already identifies whether a response contradicts the evidence, it seems plausible that it could also identify which parts of the response contain contradictions. This could enable more granular credit assignment (e.g., token-level penalties for contradictory claims) while avoiding the reward hacking issues of continuous response-level rewards.

**2. Potential Degenerate Behavior**: The proposed binary reward may implicitly incentivize shorter outputs, since fewer claims reduce the chance of contradictions. In the extreme case, a model could output an empty response and receive a reward of +1 because no contradictions are present. It would be helpful to clarify what prevents this collapse.

**3. Assumptions and Limitations**: The method assumes access to a high-quality evidence set for every query, which may not hold in many real-world settings. Additionally, the approach assumes that a verifier can reliably detect contradictions between generated responses and retrieved evidence; this may become challenging when the retrieved document set is large or when retrieval quality is imperfect. These limitations need to be acknowledged directly in the paper.

---

> ### Author Rebuttal · Authors · 2026-03-31
>
> We sincerely thank Reviewer HaLC for the thoughtful and constructive review. We appreciate the reviewer’s positive assessment of the **paper’s clarity** and the **breadth of the empirical evaluation**. Below we respond to the main concerns.
>
> - (Q1) **Convergence Speed.** Reviewer HaLC asks whether Binary RAR converges more slowly than continuous rewards, given that binary feedback provides weaker supervision.
>     - RL with Binary RAR has better final performance. Continuous rewards can reduce hallucination faster, but they yield a substantially worse utility-hallucination tradeoff. As shown in Fig. 5, under a no-utility-degradation constraint, Binary RAR achieves a much larger hallucination reduction than VeriScore.
>     - RL with Binary RAR is sufficiently efficient in practice. Our experiment uses only 400 GPU hours, compared with 17K GPU hours for general-purpose post-training of Qwen3-8B, according to the report.
> - (W1, Q2, Q4) **Dense credit assignment and Online DPO.** Reviewer HaLC suggests two related alternatives: (1) denser credit assignment that penalizes only hallucinated spans instead of the full response, and (2) Online DPO with binary preferences as a potentially more stable on-policy baseline.
>     - We additionally implemented both alternatives. Because utility on AlpacaEval already begins to drop as soon as hallucination decreases, and given the time limit, we ran these experiments on a subset of 4K prompts. Dense credit assignment reduced AlpacaEval from 41.7 to 38.4, and Online DPO reduced it from 41.7 to 36.5, whereas Binary RAR preserved utility and reached 43.0 on the full dataset (Tab. 2).
>     - For dense credit assignment, we use a Qwen3-32B verifier to assign negative rewards to tokens in sentences identified as contradictory, and then train with PPO. For Online DPO, we construct binary preference pairs and resample on-policy generations every 64 prompts. We will include this comparison and its setup more clearly in the revision.
>     - For dense credit assignment, one potential issue is that penalizing hallucinated content can still encourage the model to generate nothing in order to avoid future penalties. For Online DPO, we find that using binary preferences does not remove VeriScore’s bias, consistent with our discussion in Sec. 6.2.
>
> - (W2, Q3) **Potential Degenerate Behavior and Length Effects.** Reviewer HaLC asks whether the binary reward encourages the model to produce shorter responses in order to avoid contradictions, and whether the KL term is the only safeguard against such collapse.
>     - Our goal is not to preserve length itself, but to preserve useful content while removing hallucinated content. On the Biography dataset, the average number of claims decreases from 30.0 to 13.6 after training, while the number of correct claims remains nearly unchanged, from 8.8 to 8.6 (Tab. 3). AlpacaEval also remains essentially unchanged before and after training (54.7 vs. 53.9). This suggests that Binary RAR primarily removes hallucinated statements rather than collapsing informativeness.
>     - As discussed in Sec 3.4, Binary RAR does not inherently favor empty responses over informative contradiction-free ones. Instead, it preserves the relative distribution among contradiction-free responses and downweights contradiction-containing responses by a constant determined by the KL coefficient.
> - (W3, Q5) **Robustness to retrieval and verification.** Reviewer HaLC asks (1) how much the method depends on having high-quality retrieved evidence for each query, (2) how sensitive it is to verifier reliability when retrieval is imperfect.
>     - Our method does not require high-quality retrieved evidence. In an additional analysis of 200 randomly sampled training prompts, using responses from Qwen3-8B and GPT-5.4 web search to construct pseudo-ground-truth annotations, we found that when a response contains a factual error, the retrieval pipeline includes the relevant evidence in only 53% of cases. When evidence is missing, the verifier often cannot identify a contradiction, so different responses tend to receive the same reward and the update is close to neutral rather than systematically harmful. We will clarify this annotation protocol in the revision.
>     - In our setup, the LM verifier is sufficiently reliable. Compared against GPT-5.4-based annotations on the same 200 sampled responses, Qwen3-32B has a 16% false negative rate (a contradiction-free response is incorrectly assigned reward 0).
> - (Q6) Reviewer HaLC reports Figure 1 appears to have rendering issues; the visualization behaves weirdly when zooming in and out of the paper.
>     - Thank you for flagging this issue. We will fix it in the revision.

---

> > ### Author Rebuttal · Reviewer_HaLC · 2026-04-03
> >
> > I thank the authors for their response. Some concerns are resolved, but I still have 1 major concern:
> >
> > 1. I still do not understand why the authors chose a binary reward signal when more signal is available. In domains like Math, dense signal is generally biased (obtained using PRMs , larger models, etc) and thus using dense imperfect signal can affect performance negatively. This is one of the main reasons that standard practice is to train with binary rewards. But the setting chosen by authors is one of the rare LLM settings where it seems that perfect dense signal is available (The binary reward signal is computed by combining claim-level signals). One of the simplest reward signals I can imagine using is the fraction of correct claims for example, this already provides more signal than binary rewards. Now, authors provided an experiment where they assigned negative rewards at a token level and that does not seem to work well. This could either be because a) the method chosen to make the signal dense is suboptimal or b) dense signal is simply not fit for this setting. According to the authors, which of these 2 things is happening? Personally, I am unable to see how b) could be happening because the rewards are perfectly definable at claim level which suggests that a) is the issue, in which case more experiments would be needed. If authors can provide a concrete explanation of why issue b) can still happen, I am happy to raise my score.

---

> > > ### Author Response · Authors · 2026-04-04
> > >
> > > We thank the reviewer for the thoughtful follow-up. We appreciate the opportunity to clarify our reasoning.
> > >
> > > **Clarifying the setting.** Our goal is to balance factuality and utility in continued RL from fully trained language models. This requires a reward design that can sustain long training runs with minimal reward hacking. The reviewer suggests that a perfect dense reward should be available in our setting, since claim-level signals can in principle be aggregated. However, we find that, in practice, claim-level factuality rewards are unreliable. Noise in decomposition, retrieval, and verification makes them exploitable under RL optimization. We elaborate below.
> > >
> > > **Why do we choose response-level verification over claim-level verification?** The reviewer suggests that the fraction of correct claims would provide a simple design. Indeed, we tested exactly this (VeriScore) along with several other claim-level variants. However, all the claim-level reward formulations we tested fail to preserve utility during training. We believe this is due to the following reasons:
> > >
> > > - **Claim decomposition introduces noise.** As noted in prior work [1, 2, 3], claim decomposition is not uniquely defined. Decomposed claims frequently suffer from context omission, ambiguity, and excessive fragmentation. These issues make model ranking with claim-level metrics inconsistent across decomposition methods [1], and decompose-then-verify pipelines can sometimes even perform worse than direct verification without decomposition [2].
> > > - **RL exploits this noise.** When claim-level signals are used as rewards, RL training can exploit the noise in decomposition, retrieval, and verification rather than properly preserving utility. Our experiments in Sec. 6.2 and Fig. 4 confirm this across multiple variants. We show that specific design choices in Binary RAR, namely contradiction-only verification, response-level scoring, and binarization, are each important. We compare (1) VeriScore (fraction of correct claims), (2) Binary VeriScore (binarized VeriScore with a 0.5 threshold), (3) Rating RAR (using a 0-10 verifier score), and (4) Binary RAR.
> > >
> > > Table: Comparison of reward variants; see Fig. 4 for detailed results.
> > >
> > > |  | Biography (Hallucination Ratio,↓) | AlpacaEval (LC Win Rate,↑) |
> > > | --- | --- | --- |
> > > | Qwen3-8B | 76.2 | 54.7 |
> > > | + VeriScore (fraction of correct claims) | 51.7 | 42.2 |
> > > | + Binary VeriScore (fraction of correct claims > 0.5) | 61.3 | 48.2 |
> > > | + Rating RAR (using a 0-10 verifier score) | 65.0 | 46.9 |
> > > | + Binary RAR | 45.8 | 53.9 |
> > >
> > > - **Response-level verification is also more efficient.** By avoiding claim decomposition and reducing verification from per-claim to per-response, Binary RAR achieves 2-4x higher throughput than VeriScore. This is a practical but meaningful advantage for scaling RL-based hallucination reduction (Sec. 3.3).
> > >
> > > **Why dense token-level rewards face similar issues.** The reviewer asks whether our token-level experiment failed because (a) the attribution method we used was suboptimal or because (b) dense signals are inherently problematic here. We believe the core issue is closer to (b), for the following reasons:
> > >
> > > - **Token-level attribution of hallucination is inherently ambiguous.** Consider the example in Fig. 8: for the question "`How did US states get their names?`", the model outputs "`[...] Many states were named after British or European royalty: [...] Connecticut: Named after the English county of Connecticut.`" There are multiple valid ways to localize the error. One could attribute it to "Connecticut" appearing under the royalty category, since Connecticut was not named after British or European royalty. Alternatively, one could attribute it to "Named after the English county of Connecticut," since this gives an entirely incorrect origin. The token-level attribution depends on context and is not uniquely defined. This is unlike math reasoning, where errors can typically be localized to specific steps.
> > > - **The noise can be exploited for reward hacking.** Many incidental factors, including output length, formatting, and the use of tables, can shift how a verifier attributes errors to tokens. RL can then learn to adopt styles (e.g., shorter or more structured outputs) that make errors harder for the verifier to detect, rather than actually reducing hallucination. We discuss that the same reasons cause claim-level verification to fail to preserve utility in Sec. 6.2 and Appendix A.1.
> > >
> > > Table: Dense credit assignment is vulnerable to reward hacking.
> > >
> > > |  | AlpacaEval (LC Win Rate,↑) |
> > > | --- | --- |
> > > | Qwen3-4B | 41.7 |
> > > | + Dense Credit Assignment | 38.4 |
> > > | + Binary RAR | 43.0 |
> > >
> > > - [1] Wanner et al. (2024) A Closer Look at Claim Decomposition
> > > - [2] Hu et al. (2025) Decomposition Dilemmas: Does Claim Decomposition Boost or Burden Fact-Checking Performance?
> > > - [3] Song et al. (2024) VERISCORE: Evaluating the factuality of verifiable claims in long-form text generation

---

### Official Review · Reviewer_pZe6 · 2026-03-13

**Soundness:** 3
**Presentation:** 4
**Significance:** 3
**Originality:** 4
**Overall Recommendation:** 5
**Confidence:** 4

**Summary:**

This paper addresses the hallucination–utility trade-off in LLM post-training by proposing Binary Retrieval-Augmented Reward (Binary RAR) for online RL. The reward is binary: 1 if the model's response contains no factual contradictions with retrieved web documents, and 0 otherwise. A verifier (e.g., Qwen3-32B) checks the entire response against retrieved evidence in a single forward pass, avoiding the noise of claim-level decomposition used in continuous rewards like VeriScore. Experiments on Qwen3-4B/8B show that Binary RAR reduces long-form hallucination rates by up to 39.3% and short-form by 54.4%, while maintaining utility on general open-ended benchmarks like AlpacaEval and ArenaHard.

**Compliance With Llm Reviewing Policy:**

Affirmed.

**Final Justification:**

I thank the authors for answering my questions and for sharing their thoughts on the potential extension of Binary RAR. My concerns have been addressed. Therefore, my evaluation remains positive toward acceptance.

**Key Questions For Authors:**

- Binary RAR assigns reward 0 if any part of the response contradicts retrieved evidence. For long-form responses with many correct claims and one minor error, this seems harsh. Have you experimented with a softer threshold (e.g., reward 1 if fewer than k claims are contradicted) and compared it to the strict binary version?
- How robust is Binary RAR to the quality and coverage of the retrieval corpus? If the retrieved documents lack relevant information for a particular domain, the verifier may default to reward 0 (no evidence to support claims). Does this effectively discourage the model from generating responses for queries with poor retrieval coverage?
- Figure 2 shows that Binary RAR naturally encourages abstention on short-form QA (which can be as high as 55.2%). Have you explored mechanisms to control the abstention rate beyond adjusting the KL coefficient? For example, a ternary reward design similar in spirit to TruthRL might be worth considering: +1 if the response is supported by retrieved evidence, 0 if retrieval is uninformative or the response is "IDK", and −1 if the response contradicts the evidence. I'd be curious to hear the authors' thoughts on whether such a ternary scheme that decouples "no evidence available" from "actively wrong" could reduce over-abstention and encourage higher accuracy on topics with sparse retrieval coverage.

**Limitations:**

The authors are encouraged to discuss the limitations and potential negative societal impact of their work in the revision.

**Strengths And Weaknesses:**

Strengths
- Well-motivated reward design. Unlike VeriScore, the binary rewards are inherently resistant to reward hacking because there is no gradient to exploit through style manipulation as the reward only changes when the verifier decision flips.
- Comprehensive evaluation covering both hallucination and utility. The paper evaluates on a wide range of benchmarks and the comparison is thorough and clearly shows that both baselines and Binary RAR reduce hallucination while Binary RAR suffers the least in terms of utility degradation.
- Emergent calibrated abstention without explicit training. The finding (Figure 2) that Binary RAR training naturally increases abstention on questions the model would otherwise answer incorrectly is interesting. This emergent behavior arises from the binary reward structure without explicit abstention labels, yet achieving a similar behavior like TruthRL's ternary reward that explicitly models abstention.


Weaknesses
- The binary reward relies heavily on the verifier's accuracy, but verifier failure modes are underexplored. The verifier (Qwen3-32B) is a key component, yet its failure modes are not analyzed in depth. Section A.2 reports verifier accuracy but does not break down false positives vs. false negatives. If the verifier frequently gives false positives (incorrectly assigning reward 1 to hallucinated responses), the RL training would reinforce hallucinations. Conversely, frequent false negatives would unfairly penalize correct responses. I think a confusion matrix or error analysis of the verifier's decisions on a held-out set would further strengthen the paper.

- Limited analysis on what types of hallucinations are reduced vs. remaining. Table 2 reports aggregate hallucination rates, but it would be more informative to see a breakdown by error type (e.g., fabricated statements, numerical errors). Understanding which types of hallucinations Binary RAR is most and least effective at reducing would help practitioners decide when to apply this method. The error analysis in Section 6 discusses reward hacking during training but did not discuss the residual hallucination patterns after training.

---

> ### Author Rebuttal · Authors · 2026-03-31
>
> We sincerely thank Reviewer pZe6 for the thoughtful reviews. We appreciate the recognition of **well-motivated reward design**, the **comprehensive evaluation**, and our observation of **emergent abstention without explicit training**. We address the concerns and questions as below.
>
> - (W1) **Analysis of verifier failure modes.** Reviewer pZe6 asks for a more detailed breakdown of verifier errors, beyond the overall accuracy reported in Section A.2.
>     - To address this, we additionally analyze 200 randomly sampled training prompts with responses generated by Qwen3-8B. We then use GPT-5.4 to construct pseudo-ground truth by labeling whether each response contradicts the retrieved evidence.
>     - False negatives (where contradiction-free responses are incorrectly assigned reward 0) are especially important for utility preservation because they penalize correct responses. We find false negative rates of 16% for the Qwen3-32B verifier and 30% for the Qwen3-8B verifier, supporting our choice of the stronger verifier.
>     - False positives (where hallucinated responses are incorrectly assigned reward 1) mainly affect effectiveness of hallucination reduction because those responses are not penalized. The false positive rates are 32% for Qwen3-32B and 35% for Qwen3-8B. We agree that improving verifier reliability is an important direction for future work, and stronger verifiers could likely further improve this framework.
> - (Q1) **Soft versus strict reward thresholds.** Reviewer pZe6 asks whether the strict binary reward is too harsh for long-form responses that are mostly correct but contain a small number of errors, and whether we tested softer alternatives.
>     - We did evaluate a closely related soft-threshold variant in Section 6.3: Binary VeriScore. Under this scheme, a response receives reward 0 only when more than half of its decomposed claims are incorrect, and reward 1 otherwise. Our results show that Binary VeriScore yields worse utility than Binary RAR at comparable levels of hallucination reduction (in Fig 4). In other words, relaxing the threshold does not improve the utility-hallucination tradeoff.
>
> Table: Binary VeriScore (Figure 4)
> | Method | Hallucination (↓) | Utility (↑) |
> |---|---:|---:|
> | Binary VeriScore | 61.3 | 48.2 |
> | Binary RAR | 45.8 | 53.9 |
>
> - (Q2) **Sensitivity to retrieval quality and coverage.** Reviewer pZe6 asks whether Binary RAR remains reliable when retrieval coverage is poor, and in particular whether missing evidence would push the model to avoid answering such queries.
>     - Binary RAR is relatively robust to retrieval errors. The key reason is that it detects contradictions, rather than requiring every claim to be explicitly supported by the retrieved documents, as in attribution-based rewards such as VeriScore.
>     - If the retrieval datastore lacks relevant evidence for a query, the verifier is also less able to identify contradictions between the response and the retrieved chunks. In that case, different responses tend to receive the same reward, so the training signal is close to neutral and the model's behavior remains largely unchanged. Missing evidence therefore does not systematically discourage generation.
> - (W2) **Breakdown by hallucination type.** Reviewer pZe6 asks whether Binary RAR reduces some types of hallucinations more effectively than others, rather than only improving the aggregate rate reported in Table 2.
>     - To address this, we add a response-level error analysis on the Biography dataset. Following the reviewer's suggestion, we manually define three biography-specific hallucination categories: career errors, achievement errors, and fabricated details. For each response, we annotate whether it contains each error type, so the counts reflect the number of responses exhibiting that category.
>     - We find that Binary RAR reduces all three categories for Qwen3-8B. Responses with career errors drop from 28 to 4, achievement errors from 14 to 3, and fabricated details from 17 to 4. These results suggest that the improvement is broad and not concentrated in a single hallucination type.
> - (Q3) **Ternary reward design.** Reviewer pZe6 asks whether a ternary reward similar in spirit to TruthRL could reduce over-abstention and improve accuracy when retrieval coverage is sparse.
>     - This is a thoughtful suggestion. In our setting, we start from a fully trained model that has already been optimized for task accuracy, and then apply continual RL to reduce hallucinations while preserving that utility. This differs from TruthRL, which jointly optimizes accuracy and abstention under uncertainty. As a result, our goal is not to teach the model when to answer versus abstain, but rather to reduce hallucinated content while retaining useful content. We view ternary reward as a promising direction for future work. In particular, it would be interesting to study whether a TruthRL-style ternary reward remains effective in open-ended, long-form, chat-style settings.

---

> > ### Author Rebuttal · Reviewer_pZe6 · 2026-04-02
> >
> > I thank the authors for answering my questions and for sharing their thoughts on the potential extension of Binary RAR. My concerns have been addressed.

---

> > > ### Author Response · Authors · 2026-04-03
> > >
> > > We thank the reviewer very much for acknowledging our response. We are glad that all of your concerns have been addressed.

---

### Decision · Program_Chairs · 2026-04-30

**Decision:**

Accept (spotlight)

**Comment:**

This paper receives scores of 4, 5, 4, 5. This paper studies the hallucination-utility trade-off in post-training large language models. In the paper, the authors propose Binary Retrieval-Augmented Reward (Binary RAR), a simple online RL objective that penalizes contradiction with retrieved evidence while preserving useful capabilities. The reviewers generally agreed that the paper addresses an important problem and the paper itself is well-written. Strong empirical evidence across both factuality and utility benchmarks is also provided. In particular, the experimental results demonstrate substantial hallucination reduction together with much smaller utility degradation than continuous factuality-reward baselines, and the rebuttal further strengthened the justification for the binary, response-level design.

The discussion mainly focused on verifier reliability, retrieval dependence, and whether the gains might come from shorter or more abstaining outputs. The rebuttal addressed most of reviewers’ concerns, with additional analyses on verifier error modes, hallucination categories, dense versus binary rewards, and length effects. Several reviewers explicitly mentioned that their concerns were resolved or that they remained in favor of acceptance.

Overall, reviewers agree that this paper is technically sound, well-presented, and sufficiently significant for acceptance. The strengths outweigh the remaining concerns, especially because the paper identifies a practically useful design for hallucination mitigation and supports it with both empirical and theoretical analysis. Therefore, the recommendation is to accept this paper. For the camera-ready version, the authors are encouraged to further revise the paper by more clearly positioning the novelty relative to prior factuality-RL work, incorporating the rebuttal’s additional analyses into the main paper or appendix, and expanding the discussion of verifier reliability, retrieval limitations and abstention/length effects.